# Learning Causal Relations from Subsampled Time-Series with only Two Time-Slices

## Abstract

This paper studies the summary causal graph on subsampled time series with instantaneous effects using only two time-slices, in which time-slices (cross-sectional observations) are sampled at a coarser timescale than the causal timescale of the underlying system. Given the presence of unmeasured time-slices, conventional causal discovery methods designed for standard time series data would produce significant errors about the system's causal structure. To address these issues, one promising approach is to construct a topological ordering and then prune unnecessary edges to approximate the true causal structure. Existing topology-based methods often yield non-unique orderings with many spurious edges, reducing accuracy and efficiency in downstream search tasks, while using interventional data for more precise ordering is frequently costly, unethical, or infeasible. Therefore, we explore how the more readily available two time-slices data can replace intervention data to improve topological ordering. Based on a conditional independence criterion using two time-slices as auxiliary instrumental variables, we propose a novel Descendant Hierarchical Topology algorithm with Conditional Independence Test (DHT-CIT) to more efficiently learn causal relations in subsampled time series data. Empirical results on both synthetic and real-world datasets demonstrate the superiority of our DHT-CIT algorithm.

## 1 Introduction

Learning causal relations from time series data is a fundamental problem in many fields of science (Granger, 1969; 1980; Lütkepohl, 2005; Hyvärinen et al., 2010; Runge et al., 2019; Bussmann et al., 2021; Löwe et al., 2022; Assaad et al., 2022). While most existing methods have well-identified the causal graph from standard time series data, in many applications, the time series sampling process may be slower than the timescale of causal processes, resulting in numerous previous time-slices (cross-sectional observations) being missing or unreliable (Peters et al., 2017; Hyttinen et al., 2016). In the subsampled time series scenarios, since incorrectly modeling causal structures at the coarser timescale, conventional causal discovery methods designed for standard time series data would produce significant errors about the system's causal structure. While some recent methods can identify parts of the causal information (i.e., an equivalence class or causal ordering) from subsampled time series data, they still rely on prior knowledge of the subsampling rate and structural functions (Gong et al., 2015; Plis et al., 2015; Hyttinen et al., 2016). Therefore, learning causal structure from subsampled time series (Fig. 1(a,b))is still challenging without efficient solutions yet.

Inspired by extension works (Runge et al., 2019; Malinsky & Spirtes, 2018; Hyvärinen et al., 2010), a naive thought is to directly extend conventional non-temporal methods to the subsampled time-series setting, incorporating additional assumptions. However, this remains challenging due to the super-exponential search space and the lack of guarantees for identifiable results. To address these issues, one more promising approach is to construct a topological ordering and then prune unnecessary edges to approximate the true causal structure (Teyssier & Koller, 2005; Peters et al., 2014; Loh & Bühlmann, 2014; Park & Klabjan, 2017). For example, SCORE and DiffAN (Rolland et al., 2022; Sanchez et al., 2022) use the Hessian of the data log-likelihood to iteratively identify leaf nodes and generate a complete topological ordering (Fig. 1(c)) to approximate the true causal structure. However, these methods typically generate non-unique topological ordering with numerous spurious edges, which poses potential difficulties in downstream search tasks and may lead to errors in learning summary causal graphs (Fig. 1(a) (Rolland et al., 2022; Sanchez et al., 2022)).

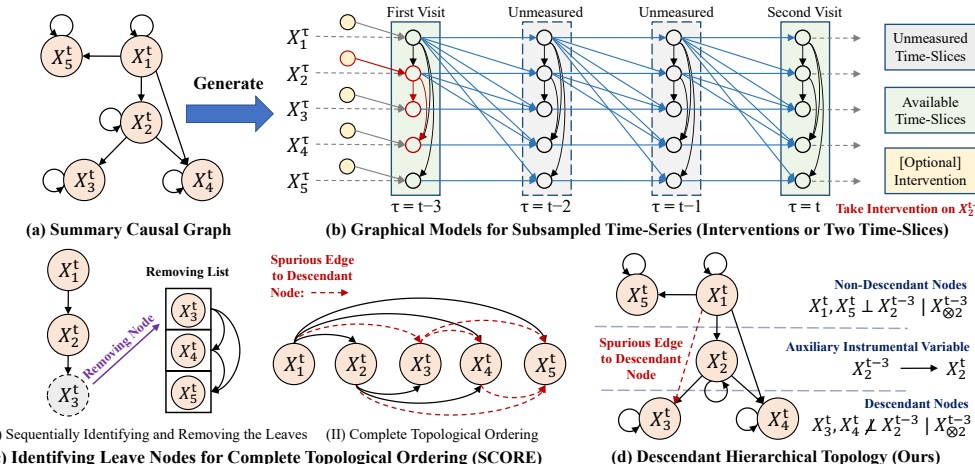

Figure 1: (a, b) Subsampled Time Series with Only Two Time-Slices. (c, d) The SCORE and DHT-CIT architectures. In healthcare, doctors typically compare earlier $X^{t-3}$ and current $X^t$ patient records to identify causes of outcome of interest. Patient visits may be recorded less frequently than the causal timescale of the underlying system, leaving $X^{t-2}$ and $X^{t-1}$ unrecorded.

To construct more precise topological orderings for learning DAGs of Summary Causal Graph, a typical solution is to conduct intervention experiments on each node and then make a comparison to identify which variables have been influenced by the intervention. As marked in red in Fig. 1(b), intervening on $X_2^{t-3}$ enables quick identification of its descendants ($X_3^{t-3}$, $X_4^{t-3}$) and improves the topological ordering. However, full interventions are often expensive, unethical, or even infeasible (Wang et al., 2017; Yang et al., 2018). Therefore, in the subsampled time series setting, we explore using just two time-slices to replace interventions to improve topological ordering, in which a time-slice denotes sampled cross-sectional observations on multiple samples captured at a time point on the timeline of system. As illustrated in Fig. 1(b), where each node is influenced by its ancestors and itself, each variable in the earlier time-slice would transmit self-perturbations to both itself and its descendants in the subsequent time-slice, serving as simulated interventions.

Unlike traditional time series studies where all previous time-slices within the observation windows are accessible, in many subsampled time series scenarios, we have access only to limited reliable time-slices. This situation is quite common in healthcare, as shown in Fig. 1(b), where doctors typically use limited time-slices to analyze a patient's condition and determine treatments. Therefore, in this paper, we study the summary causal graph on subsampled time series with instantaneous effects using only two time-slices. Based on a conditional independence criterion using the earlier time-slice as auxiliary instrumental variables, we propose a Descendant Hierarchical Topology algorithm with Conditional Independence Test (DHT-CIT) to quickly identify (non-)descendants for each node (Fig. 1(d)), ultimately constructing a more efficient unique Descendant Hierarchical Topology with merely a few spurious edges. Subsequently, we prune unnecessary edges to approximate the true causal graph. We rigorously prove that it is both sound and complete under some graph constraints.

The contribution of our paper is four-fold:

- We study the Directed Summary Causal Graph on Subsampled Time Series with Instantaneous Effects using only two time-slices, in which time-slices (cross-sectional observations) are sampled at a coarser timescale than the causal timescale of the underlying system.

- Theoretically, we prove that by incorporating a previous state or implementing interventions for each node, a single conditional independence test per node is sufficient to distinguish between its descendants and non-descendant nodes in the topological ordering.

- Methodologically, we propose DHT-CIT, a novel identifiable topology-based algorithm for a unique descendant hierarchical topology. The search space for underlying graphs over the learned descendant hierarchical topology is much smaller than conventional methods.

- Empirically, our algorithm demonstrates superior performance in synthetic data experiments and partially identifies causal relationships in a real-world study on cardiovascular mortality rate (CMR), providing valuable insights for personalized policy decisions.

## 2  RELATED WORK ON TEMPORAL DATA

Standard methods for inferring causal structure from conventional time series typically focus either on estimating a transition model at the measurement timescale (e.g., Granger causality (Granger, 1969; 1980)) or they integrate a model of measurement timescale with 'instantaneous' or 'contemporaneous' causal relations to capture interactions within and between d-variate time series from observational data (Lütkepohl, 2005; Hyvärinen et al., 2010; Luo et al., 2015; Nauta et al., 2019; Runge et al., 2019; Runge, 2020; Bussmann et al., 2021; Löwe et al., 2022; Assaad et al., 2022). However, these methods depend on modeling causal structures at the system timescale and assuming causal sufficiency. Both of these conditions might not hold in Subsampled Time Series with only two time-slices, as there could be numerous unmeasured time slices latent in the time series, either before or between these two observed time-slices (Peters et al., 2017).

Subsampled processes with a few time-slices in time series setting are ubiquitous and inherent in the real world, however, causal discovery over Subsampled Time Series is not as well explored. With the prior of the degree of undersampling, (Gong et al., 2015) uses Expectation-Maximization algorithm to recover the linear temporal causal relations from the subsampled data. (Tank et al., 2019) take structural vector autoregressive models for parameter identifiability and estimation. The identifiability of both works is achieved only for linear data. As for nonlinear data, (Danks & Plis, 2013) only can extract constraints on summary graphs from the strongly connected components (SCC). Inspired by works (Gong et al., 2015; Plis et al., 2015), (Hyttinen et al., 2016) proposes a constraint optimization approach to identify a small part of the causal information (i.e., an equivalence class) from subsampled time series data, but requires no instantaneous effects. Causal discovery from subsampled time series is still challenging without an efficient solution yet.

Recently, promising topology-based methods tackle the causal discovery problem by finding a certain topological ordering of the nodes and then pruning the spurious edges (Teyssier & Koller, 2005; Peters et al., 2014; Loh & Bühlmann, 2014; Park & Klabjan, 2017; Ghoshal & Honorio, 2018; Ahammad et al., 2021; Sanchez et al., 2022; Reisach et al., 2023). In this paper, we study the Directed Summary Causal Graph on Subsampled Time Series with instantaneous effects using only two time-slices and explore using two time-slices as a substitute for intervention data to improve causal ordering of Summary Causal Graph in Subsampled Time Series. More related works about non-temporal data and their variants can be found in Appendix A and C.

## 3  PROBLEM SETUP

**Standard Time Series:** Let $\boldsymbol{X} = \{X_i^\tau\}_{d \times t}$ denote the full $d$-variate time series with all time slices $\boldsymbol{X}^\tau$ at $t$ time points, where $X_i^\tau$, $i \in \{1, 2, \cdots, d\}$ and $\tau \in \{1, 2, \cdots, t\}$, is a random vector comprising observations of $n$ samples. For simplicity of notation, we will not discuss each sample individually. Instead, we refer to $X_i^\tau$ as a random variable when discussing the causal structure. The true causal structure of the summary causal graph is represented by a DAG $\mathcal{G}$. For each $X_i^\tau$, we use the notation $\mathbf{pa}_i^\tau$ to refer to the set of parents of $X_i^\tau$ at time $\tau$ in $\mathcal{G}$. Similarity, we define $\mathbf{ch}_i^\tau$ for the set of child nodes, $\mathbf{an}_i^\tau$ for the ancestors set, $\mathbf{sib}_i^\tau$ for the siblings set, and $\mathbf{de}_i^\tau$ for the descendants set. As shown in Fig. 1(a,b), following the summary causal graph, the causal structure can be expressed in the functional relationship, for $i \in \{1, 2, \cdots, d\}$ and $\tau \in \{1, 2, \cdots, t\}$:

$$X_i^\tau = f_i \left( \mathbf{pa}_i^\tau, X_i^{\tau-1}, \mathbf{pa}_i^{\tau-1} \right) + \epsilon_i^\tau, \tag{1}$$

where $f_i(\cdot)$ is a twice continuously differentiable function capturing the instantaneous effects from its parents $\mathbf{pa}_i^\tau$ at time $\tau$ and a non-zero time-lagged effect from variable $X_i^{\tau-1}$ at time $\tau - 1$; and $\epsilon_i^\tau$ is an *Additive Noise* at time-slice $\tau$.

**Subsampled Time Series with Two Time-Slices:**. In this paper, we focus on learning the DAG $\mathcal{G}$ of the summary causal graph (Fig. 1(a)) on subsampled time series (Fig. 1(b)) with instantaneous effects using only two time-slices $\mathcal{D} = \{\boldsymbol{X}^{t_a}, \boldsymbol{X}^{t_b}\}_{1 < t_a < t_b < t}$. Based on a conditional independence criterion using earlier time-slice $\boldsymbol{X}^{t_a}$ as conditional instrumental variables[1], we can easily

---

[1] Given $\mathbf{pa}_i^\tau$, the auxiliary variable $X_i^\tau$ (i.e., conditional instruments) is conditional independent with its non-descendants $\{\mathbf{an}_i^\tau, \mathbf{sib}_i^\tau\}$ and only indirectly affects its descendant nodes $\mathbf{de}_i^\tau$ (i.e., outcomes) at the current state through its association with $X_i^t$ (i.e., treatments).

distinguish descendants and non-descendants for each node in the current time-slice $\boldsymbol{X}^{t_b}$. We rigorously prove that it is both sound and complete under some graph constraints, i.e., Markov property, acyclic summary causal graph, stationary full-time graph. More discussion about advantages and limitations of using two time-slices data could be found in Appendix G.

**Assumption 1** (Markov Property). *The Markov property of a time series assumes the future time-slice $\boldsymbol{X}^{t+1}$ depends only on current state $\boldsymbol{X}^t$ and does not depend on past history $\boldsymbol{X}^{1\cdots t-1}$.*

**Assumption 2** (Acyclic Summary Causal Graph, Section 5.2.1 in (Assaad et al., 2022))). *The summary causal graph of a multivariate time series is considered acyclic if the lagged effect of each variable solely affects its own value and its descendants, without any influence on its non-descendants.*

**Assumption 3** (Consistency Throughout Time, Definition 7 in (Assaad et al., 2022))). *A causal graph $\mathcal{G}$ for a multivariate time series $\boldsymbol{X}$ is said to be consistent throughout time if all the causal relationships remain constant throughout time, also referred to as stationary full-time graph.*

Following these assumptions, the topological ordering is known to be identifiable from observational data (Peters et al., 2014; Bühlmann et al., 2014), and it is possible to recover the DAG of summary graph underlying the additive noise models (Eq. equation 1). Furthermore, in Appendix F, we relax the Markov Assumption to a high-order Markov Assumption and extend our algorithm to subsampled time series with high-order lagged effect.

## 4 ALGORITHM

In this section, we will first introduce the complete topological ordering from classical topology-based approaches (Rolland et al., 2022; Sanchez et al., 2022) and show how two time-slice data help identify a unique descent hierarchical topology. Then, based on a conditional independence criterion using the previous time-slice as auxiliary instrumental variables, we propose a novel identifiable topology-based algorithm (DHT-CIT) for two time-slices, which is applicable to any type of noise. The search space over the learned descendant hierarchical topology is much smaller than that of advanced approaches. Then, the underlying summary graph can be found by pruning the unnecessary edges with a well-defined pruning method (Bühlmann et al., 2014).

### 4.1 FROM COMPLETE TO DESCENDANT HIERARCHICAL TOPOLOGY

As shown in Fig. 1(c), the conventional typology-based approach SCORE (Rolland et al., 2022; Sanchez et al., 2022) sequentially identifies and removes leaf nodes to generate a complete topological ordering based on the Hessian's diagonal of the data log-likelihood.

**Definition 1** (Complete Topological Ordering). *The complete topological ordering ($\pi(\boldsymbol{X}) = (X_{\pi_1}, X_{\pi_2}, \cdots, X_{\pi_d})$, $\pi_i$ is the reordered index of node) is a sorting of all nodes in a DAG such that for any pair of nodes $X_{\pi_i}$ and $X_{\pi_j}$, if there exists a directed edge from $X_{\pi_i}$ to $X_{\pi_j}$, then $i > j$.*

However, a complete topological ordering is a dense graph with $d(d-1)/2$ edges, containing numerous spurious edges, many of which point to non-descendants unnecessarily. Moreover, these methods (Rolland et al., 2022; Sanchez et al., 2022) may not always produce a unique solution, making it challenging to eliminate false edges and resulting in errors when learning summary causal graph. Fortunately, obtaining two time-slices data can help identify a unique hierarchical topological ordering (Fig. 1(d)), i.e., Descendant Hierarchical Topology, in which each edge only points from an ancestor node to its descendant nodes and not to any non-descendant nodes.

**Definition 2** (Hierarchical Topological Ordering). *In the hierarchical topological ordering e.g., $\Pi(\boldsymbol{X}) = (\{X_{\pi_1}\}_{\boldsymbol{L}_1}, \{X_{\pi_2}, X_{\pi_3}\}_{\boldsymbol{L}_2}, \cdots)$, each layer is denoted by $\boldsymbol{L}_i$ and the located layer of $X_j$ are represented as $l_j$. If there is a directed edge from $X_{\pi_i}$ to $X_{\pi_j}$, then $l_{\pi_i} > l_{\pi_j}$.*

**Definition 3** (Descendant Hierarchical Topology). *In the descendant hierarchical topology, each node $X_i^t$ identifies other nodes as either non-descendant nodes or descendant nodes, and each node $X_i^t$ establishes direct edges pointing to its descendant nodes $\mathbf{de}_i^t$, i.e., $X_i^t \rightarrow \mathbf{de}_i^t, i \in \{1, 2, \cdots, d\}$.*

For a given causal graph, there may be multiple complete topological orderings (CTO) and hierarchical topological orderings (HTO). However, the descendant hierarchical topology is unique and contains fewer non-essential edges compared to CTO and HTO. This improvement eliminates spurious edges pointing to non-descendant nodes in the learned Descendant Hierarchical Topology and reduces the search space during the pruning stage of typology-based methods.

## 4.2 Conditional Independence Criterion Arising from Two Time-Slices

Two time-slices help topological ordering for learning summary causal graphs. Based on a conditional independence criterion using previous time-slice $\boldsymbol{X}^{t_a}$ as auxiliary instrumental variables, we can easily distinguish descendants and non-descendants for each node in the current time-slice $\boldsymbol{X}^{t_b}$.

**Theorem 1** (Descendant-oriented Conditional Independence Criteria). *Given two time-slice observations $\mathcal{D} = \{\boldsymbol{X}^{t_a}, \boldsymbol{X}^{t_b}\}_{t_a < t_b}$ satisfying Assumptions 1, 2, and 3, for variables $X_i^{t_a}$ and $X_i^{t_b}$, where $i \in \{1, 2, \cdots, d\}$, we can conclude that $X_j^{t_b}$ is a descendant node of $X_i^{t_b}$ iff $X_i^{t_a} \not\perp X_j^{t_b} \mid \mathbf{an}_i^{t_a}$.*

*Proof.* According to the non-zero *time-lagged effect* of $X_i^{t_a}$ on $X_i^{t_b}$ (Eq. (1)), we have the causal path: **(a)** $X_i^{t_a} \dashrightarrow X_i^{t_b}$. Under Assumption 1, we have: **(b)** $\boldsymbol{X}^{\tau} \not\rightarrow X_i^{t_b}$ for all time-slices $\tau < t_a < t_b$. Under Assumption 2, we have: **(c)** $X_i^{t_a} \not\rightarrow \mathbf{an}_i^{t_b}$ for $t_a < t_b$. Under Assumption 3, we observe a path: **(d)** $X_i^{t_a} \dashrightarrow X_j^{t_a} \dashrightarrow X_j^{t_b}$ Given (a), (b), (c) and (d), if $X_j^{t_b} \in \mathbf{an}_i^{t_b}$, then there are only two summary paths between $X_i^{t_a}$ and $X_j^{t_b}$: $X_i^{t_a} \leftarrow\!\!- \mathbf{an}_i^{t_a} \dashrightarrow X_i^{t_b}$ and $X_i^{t_a} \dashrightarrow \{X_i^{t_b}, \mathbf{de}_i^{t_b}\} \leftarrow\!\!- X_j^{t_b}$. Hence, once we control the conditional set $\mathbf{an}_i^{t_a}$, i.e., cut off all backdoor path, then the confounding effect between $X_i^{t_a}$ and $X_j^{t_b}$ would be eliminated and $X_i^{t_a} \perp\!\!\!\perp X_j^{t_b} \mid \mathbf{an}_i^{t_a}$. Similarity, if $X_j^{t_b} \in \mathbf{sib}_i^{t_b}$, then the summary backdoor path is $X_i^{t_a} \leftarrow\!\!- \mathbf{an}_i^{t_a} \dashrightarrow \mathbf{an}_j^{t_b} \dashrightarrow X_j^{t_b}$. In summary, if $X_j^{t_b}$ is a non-descendant node of $X_i^{t_b}$, then $X_i^{t_a} \perp\!\!\!\perp X_j^{t_b} \mid \mathbf{an}_i^{t_a}$. In turn, given the condition $X_i^{t_a} \not\perp X_j^{t_b} \mid \mathbf{an}_i^{t_a}$, $X_j^{t_b}$ is a descendant node of $X_i^{t_b}$. $\qquad\square$

However, since the causal graph is unknown, we are unable to directly determine the ancestor nodes $\mathbf{an}_i^{t_a}$. Therefore, we choose $\boldsymbol{X}_{\otimes i}^{t_a}$ to include all variables at time $\tau$, except for $X_i^{t_a}$ and any variables that are independent of $X_i^{t_a}$, as the conditional set. This means that $X_i^{t_a} \not\perp X_j^{t_a}$ for each variable $X_j^{t_a} \in \boldsymbol{X}_{\otimes i}^{t_a}$. As $\boldsymbol{X}_{\otimes i}^{t_a}$ occurs prior to time $t$, it does not introduce any additional backdoor paths to non-descendant nodes at time $t$, nor can it block the path $X_i^{t_a} \dashrightarrow X_i^{t_b} \dashrightarrow X_j^{t_b}$. Thus, we can directly modify the conditional set in Theorem 1 to $\boldsymbol{X}_{\otimes i}^{t_a}$.

**Corollary 1.** *Given a two time-slice observations $\mathcal{D} = \{\boldsymbol{X}^{t_a}, \boldsymbol{X}^{t_b}\}_{\tau < t}$, for the variables $X_i^{t_a}$ and $X_i^{t_b}$, where $i = 1, 2, \cdots, d$, we have $X_j^{t_b}$ is a descendant node of $X_i^{t_b}$ iff $X_i^{t_a} \not\perp X_j^{t_b} \mid \boldsymbol{X}_{\otimes i}^{t_a}$.*

Based on the corollary 1, we can distinguish between descendant and non-descendant nodes of each variable $X_i$ by conducting a single conditional independence test per variable ($X_i^{t_a} \not\perp \mathbf{de}_i^{t_b} \mid \boldsymbol{X}_{\otimes i}^{t_a}$). Additionally, if we perform a random intervention on $X_i^{t_a}$, the conditional set in corollary 1 will be empty because the random intervention is independent of the other variables. As a result, the conditional independence test in corollary 1 can be replaced with a simple independence test, which will effectively speed up the search for descendant hierarchical topology. The difference between the theorems of this paper with those of traditional methods are placed in Appendix B.

## 4.3 DHT-CIT Algorithm

### 4.3.1 Identifying descendant hierarchical topology

Based on the Conditional Independence Criteria in Corollary 1 and using previous time-slice $\boldsymbol{X}_{\otimes i}^{t_a} = \{X_j^{t_a} \mid X_j^{t_a} \perp\!\!\!\perp X_i^{t_a}\}$ as AIVs, we can identify descendants $\mathbf{de}_i^{t_b}$ of each variable $X_i^{t_a}$ by a single conditional independence test per variable ($X_i^{t_a} \not\perp \mathbf{de}_i^{t_b} \mid \boldsymbol{X}_{\otimes i}^{t_a}$). For every $i, j \in \{1, 2, \cdots, d\}$, we calculate the conditional independence significance $\boldsymbol{P}$ using the conditional HSIC test from (Zhang et al., 2011) with Gaussian kernel. We determine that $X_i$ is a descendant of $X_j$ if the reported $p$-value is less than or equal to a threshold $\alpha$, i.e., $X_i^{t_a} \not\perp X_j^{t_b} \mid \boldsymbol{X}_{\otimes i}^{t_a}$. Then, we can obtain the adjacency matrix of the unique descendant hierarchical topology by:

$$\boldsymbol{P} = \begin{pmatrix} p_{1,1} & p_{1,2} & \cdots & p_{1,d} \\ p_{2,1} & p_{2,2} & \cdots & p_{2,d} \\ \vdots & \vdots & \ddots & \vdots \\ p_{d,1} & p_{d,2} & \cdots & p_{d,d} \end{pmatrix}, \boldsymbol{A}^{TP} = \begin{pmatrix} \mathbb{I}(p_{1,1} \leq \alpha) & \mathbb{I}(p_{1,2} \leq \alpha) & \cdots & \mathbb{I}(p_{1,d} \leq \alpha) \\ \mathbb{I}(p_{2,1} \leq \alpha) & \mathbb{I}(p_{2,2} \leq \alpha) & \cdots & \mathbb{I}(p_{2,d} \leq \alpha) \\ \vdots & \vdots & \ddots & \vdots \\ \mathbb{I}(p_{d,1} \leq \alpha) & \mathbb{I}(p_{d,2} \leq \alpha) & \cdots & \mathbb{I}(p_{d,d} \leq \alpha) \end{pmatrix}. \quad (2)$$

where $p_{i,j} = \mathbf{HSIC}(X_i^{t_a}, X_j^{t_b} \mid \boldsymbol{X}_{\otimes i}^{t_a})$, $\alpha$ is a hyper-parameter denoting significance threshold, and $\mathbb{I}(\cdot)$ is the indicator function. If the $p$-value is less than $\alpha$, the result is considered significant and an

edge is added in the descendant hierarchical topology. In statistical hypothesis testing, $\alpha$ is typically set to 0.05 or 0.01. In this paper, we set the hyper-parameter $\alpha = 0.01$ as the default.

Despite the significant advancements in the development of conditional independence (Zhang et al., 2011; Runge, 2018; Bellot & van der Schaar, 2019), testing for conditional independence remains a challenging task, particularly in high-dimensional cases. The conditional independence test (HSIC) may result in biased topological ordering with cycles. To address this issue, we propose topological layer adjustment as a double guarantee for acyclic constraints in causal discovery.

### 4.3.2 Adjusting the topological ordering

To correct the conditional independence test and avoid cycles in the topological ordering, we propose topological layer adjustment to rectify the cycle graph in ordering.

**Identifying Leaf Layer**. We systematically identify the leaf nodes of the descendant hierarchical topology layer by layer, specifically by selecting nodes that do not have any descendant nodes. We denote all variables at time-slice $\tau$, except for $X_i^\tau$, as $\boldsymbol{X}_{-i}^\tau$, where $\tau \in \{1, 2, \cdots, t\}$. At $k$-th leaf layer $\boldsymbol{L}_k$, if $X_i^{t_b} \perp \boldsymbol{X}_{-i}^{t_a} \mid \boldsymbol{X}_{\otimes i}^{t_a}$, then $X_i^{t_b}$ is a leaf node at time-slice $\tau = t_b$ and $X_i \in \boldsymbol{L}_k$. By repeating this operation, we can iteratively $k := k + 1$ and identify the current leaf layer $\boldsymbol{L}_k$:

$$X_i^{t_b} \in \boldsymbol{L}_k, \text{ if } a_{i,j}^{TP} = 0 \text{ for all } j \in M_{i,k}, \tag{3}$$

where $X_{M_{i,k}} = \{\boldsymbol{X}^{t_a}/X_i^{t_a}, \boldsymbol{L}_{1:k-1}\}$ denotes all variables at time-slice $t_a$, except for $X_i^{t_a}$ and the variables in lower layer $\boldsymbol{L}_{1:k-1}$. Then $M_{i,k}$ is the index of variables $\{\boldsymbol{X}^{t_a}/X_i^{t_a}, \boldsymbol{L}_{1:k-1}\}$.

**Ensuring Acyclic Constraints**. By repeating the above procedure, we can sequentially leaf nodes layer-by-layer until we encounter cycles in the topological ordering, which makes it impossible to identify any leaf node as all nodes have at least one descendant node at this time. To ensure acyclic constraints and rectify the edges in descendant hierarchical topology, if the causal relationship between the unprocessed nodes in topological ordering forms a DAG, we locate the maximum $p$-value that is less than $\alpha$ and change it to a value of $2\alpha$, deleting the corresponding edge in the topology

$$p_{i^*,j^*} := 2\alpha \quad \text{and} \quad a_{i^*,j^*}^{TP} = 0, \quad (i^*, j^*) = \arg\max_{i,j}(p_{i,j} \leq \alpha), \tag{4}$$

we repeat this operation until a new leaf node is identified. By adjusting the $p$-value, the layer sorting leads to a more precise hierarchical topological ordering $\boldsymbol{A}^{TP} = \{a_{i,j}^{TP}\}_{d \times d}$. This ensures that the topological ordering of the graph is acyclic and improves the accuracy of topological ordering.

### 4.3.3 Pruning spurious edges

Based on a conditional independence criterion and two time-slices data, we propose a DHT-CIT algorithm to construct a more efficient descendant hierarchical topology with merely a few spurious edges (Fig. 1(d)). Theoretically, the conditional independence between the hierarchical topological layer ordering allows for a pruning process that only requires one higher layer's nodes, current layers' nodes and two lower layers' nodes as the conditional set, or the node's non-descendants and one lower layer's nodes as the conditional set, to examine if a spurious edge exists between a node and its descendant nodes. However, classical methods such as CAM appear to perform better in practice (Bühlmann et al., 2014), which use significance testing based on generalized additive models and select cause if the $p$-values are less than or equal to 0.001. Like (Rolland et al., 2022), we use CAM to prune spurious edges. The pseudo-code is placed in Algorithm 1 in Appendix D.

## 5 Numerical experiments

### 5.1 Baselines and evaluation

In the experiments, we provide a broad range of time series variants of conventional non-temporal methods that utilize a concatenation of the two cross-sectional data and the temporal edge as additional information to initialize the adjacency matrix and remove temporal edges that are not the same variable. Then, we apply the proposed algorithm (**DHT-CIT**) to both synthetic and real-world data and compare its performance to the following baselines: constraint-based methods, **PC**

and **FCI** (Spirtes et al., 2000); score-based methods, **GOLEM** (Ng et al., 2020), **NOTEARS** with MLP (Zheng et al., 2020), and **ReScore** (Zhang et al., 2023); traditional time-seires method, Granger (Shojaie & Michailidis, 2010), VARLiNGAM (Hyvärinen et al., 2010), and **CD-NOD** (Huang et al., 2020); topology-based methods, **CAM** (Bühlmann et al., 2014) and **SCORE** (Rolland et al., 2022). The discussions about the rationale behind the chosen baselines are deferred to Appendix C.

To evaluate the performance of the proposed **DHT-CIT**, we compute the Structural Hamming Distance (**SHD**) between the output and the true graphs, which evaluates the differences in terms of node, edge, and connection counts in the two graphs. Besides, we use Structural Intervention Distance (**SID**) to count the minimum number of interventions required to transform the output DAG into the true DAG, or vice versa. The accuracy of the identified edges can also be evaluated through the use of commonly adopted metrics **F1-Score** and L2-distance (**Dis.**) between two graphs.

In topology-based methods, SCORE often generates a complete topological ordering with $d(d-1)/2$ edges. However, many of these edges are unnecessary and point to non-descendants, requiring pruning, which can decrease the accuracy and efficiency of the pruning process. The proposed DHT-CIT improves upon existing typology-based methods by eliminating numerous spurious edges in the learned Descendant Hierarchical Topology and reducing the search space in the pruning stage As a comparison among topology-based methods, we count the number of spurious edges that needed to be pruned for each method, which is represented by **#Prune**.

## 5.2 EXPERIMENTS ON SYNTHETIC DATA

**Datasets**. We test our algorithm on synthetic data generated from a *additive non-linear noise model* (Eq. 1) under Assumptions 1, 2 and 3. Given $d$ nodes and $e$ edges, we generate the causal graph $\mathcal{G}$ using Erdos-Renyi model (Erdös & Rényi, 2011). In main experiments, we generate the data with Gaussian Noise for every variable $X_i^\tau$, $i = 1, 2, \cdots, d$ at time $\tau = 1, 2, \cdots, t$:

$$X_i^\tau = \text{Sin}\left(\mathbf{pa}_i^\tau, X_i^{\tau-1}\right) + \tfrac{1}{10}\text{Sin}\left(\mathbf{w} \cdot \mathbf{pa}_i^{\tau-1}\right) + \epsilon_i^\tau, \mathbf{X}^0 \sim \mathcal{N}\left(0, \text{I}_d\right), \boldsymbol{\epsilon}^\tau \sim \mathcal{N}\left(0, 0.4 \cdot \text{I}_d\right), \quad (5)$$

where $\text{Sin}(\mathbf{pa}_i^\tau) = \sum_{j \in \text{pa}(X_i)} \sin(X_j^\tau)$, $\text{I}_d$ is a $d$-th order identity matrix, and $\mathbf{w}$ is a random 0-1 vector that controls the number and existence of time-lagged edges from $\mathbf{pa}_i^{\tau-1}$. The experiments on varying time-lagged edges are deferred to Fig. 2, while in the main experiments, we set the number of time-lagged edges from $\mathbf{pa}_i^{\tau-1}$ as 0. To evaluate the performance of our DHT-CIT across various scenarios, we vary the number of nodes ($d$) and edges ($e$) to generate larger and denser graphs, which we refer to as **Sin-$d$-$e$**. Moreover, to test the algorithm's robustness against different noise types, we also generate data with Laplace noise ($X_i^0, \epsilon^\tau \sim \text{Laplace}(0, 1/\sqrt{2})$) and Uniform noise ($X_i^0, \epsilon^\tau \sim U(-1, 1)$). In each experiment setting, we perform 10 replications, each with a sample size 1000, to report the mean and the standard deviation of the mentioned metrics.

Additional, experiments on exploring **complex non-linear relationships** and on **large graph with 50/100-dimension variables** are deferred to Appendix E.1 and E.2.

**Investigating the Influence of Underlying DAG's Size and Sparsity in Sin-$d$-$e$ Experiments**. The results of the synthetic experiments are shown in Tab. 1 and Fig. 2. From the results on sparse graphs (**Sin-10-10** and **Sin-20-20**) in Tab. 1, we have the following observation: (1) In non-linear time series data, causal sufficiency and time dependency can be violated. Despite concatenating two time-slices, the time series variants of PC and FCI still fail to identify causal graphs. (2) The three methods (GOLEM, NOTEARS, and ReScore) specifically designed for sparse graphs have shown excellent performance, surpassing the proposed DHT-CIT on Sin-10-10 with observational data ($\mathcal{D} = \{\mathbf{X}^1, \mathbf{X}^2\}$). However, their performance significantly deteriorates on denser graphs (Fig. 2). (3) Traditional time series algorithms like Granger and VARLiNGAM, which rely on modeling at the standard measurement timescale, struggle with identifying the summary causal graph of subsampled time series. They even underperform compared to temporal variants of conventional methods designed for non-temporal data. While the CD-NOD method shows promising results in small-sized datasets, it only offers an equivalence class of the causal graph, which constrains the exploration of true causality. (4) As a topology-based method, SCORE was able to recover nearly true causal structure when applied to interventional data ($\mathcal{D} = \{\mathbf{X}^0, \mathbf{X}^1\}$). However, its performance decreases when dealing with observational data ($\mathcal{D} = \{\mathbf{X}^1, \mathbf{X}^2\}$) due to the presence of complex causal relationships from previous states. (5) The proposed DHT-CIT builds a descendant hierarchical topology with merely a few spurious edges. The search space over the learned descen-

Table 1: The results (mean$_{\pm std}$) on sparse graph Sin-$d$-$e$ with simulated interventional data ($\mathcal{D} = \{\boldsymbol{X}^0, \boldsymbol{X}^1\}$) or observational data ($\mathcal{D} = \{\boldsymbol{X}^1, \boldsymbol{X}^2\}$).

| Method | Sin-10-10 Graph with Interventional Data ($\mathcal{D} = \{\boldsymbol{X}^0, \boldsymbol{X}^1\}$) | | | | | Sin-10-10 Graph with Observational Data ($\mathcal{D} = \{\boldsymbol{X}^1, \boldsymbol{X}^2\}$) | | | | |
|---|---|---|---|---|---|---|---|---|---|---|
| | SHD↓ | SID↓ | F1-Score↑ | Dis.↓ | #Prune↓ | SHD↓ | SID↓ | F1-Score↑ | Dis.↓ | #Prune↓ |
| PC | $5.90_{\pm3.28}$ | $34.7_{\pm20.8}$ | $0.77_{\pm0.11}$ | $2.32_{\pm0.74}$ | - | $12.8_{\pm5.03}$ | $43.6_{\pm9.94}$ | $0.56_{\pm0.12}$ | $3.51_{\pm0.72}$ | - |
| FCI | $9.70_{\pm2.87}$ | $58.9_{\pm17.3}$ | $0.67_{\pm0.07}$ | $3.08_{\pm0.48}$ | - | $15.3_{\pm3.77}$ | $71.0_{\pm11.5}$ | $0.54_{\pm0.09}$ | $3.89_{\pm0.46}$ | - |
| GOLEM | $\mathbf{0.00}_{\pm0.00}$ | $\mathbf{0.00}_{\pm0.00}$ | $\mathbf{1.00}_{\pm0.00}$ | $\mathbf{0.00}_{\pm0.00}$ | - | $\mathbf{0.50}_{\pm0.80}$ | $1.80_{\pm2.70}$ | $\mathbf{0.97}_{\pm0.03}$ | $\mathbf{0.38}_{\pm0.59}$ | - |
| NOTEARS | $\mathbf{0.00}_{\pm0.00}$ | $\mathbf{0.00}_{\pm0.00}$ | $\mathbf{1.00}_{\pm0.60}$ | $\mathbf{0.00}_{\pm0.00}$ | - | $1.20_{\pm0.60}$ | $2.30_{\pm1.20}$ | $0.94_{\pm0.02}$ | $1.02_{\pm0.30}$ | - |
| ReScore | $\mathbf{0.00}_{\pm0.00}$ | $\mathbf{0.00}_{\pm0.00}$ | $\mathbf{1.00}_{\pm0.00}$ | $\mathbf{0.00}_{\pm0.00}$ | - | $1.00_{\pm0.63}$ | $\mathbf{1.40}_{\pm1.36}$ | $0.95_{\pm0.03}$ | $0.88_{\pm0.47}$ | - |
| Granger | $26.5_{\pm5.22}$ | $75.6_{\pm13.8}$ | $0.20_{\pm0.08}$ | $5.12_{\pm0.52}$ | - | $31.3_{\pm11.6}$ | $66.8_{\pm30.8}$ | $0.21_{\pm0.04}$ | $5.48_{\pm1.10}$ | - |
| VarLiNGAM | $35.0_{\pm0.00}$ | $69.4_{\pm3.20}$ | $0.36_{\pm0.00}$ | $5.91_{\pm0.00}$ | - | $35.0_{\pm0.00}$ | $69.4_{\pm3.20}$ | $0.36_{\pm0.00}$ | $5.91_{\pm0.00}$ | - |
| CD-NOD | $3.00_{\pm3.16}$ | $11.3_{\pm13.3}$ | $0.86_{\pm0.14}$ | $1.28_{\pm1.16}$ | - | $5.40_{\pm0.92}$ | $15.5_{\pm4.70}$ | $0.74_{\pm0.04}$ | $2.32_{\pm0.19}$ | - |
| CAM | $5.00_{\pm6.27}$ | $14.9_{\pm18.5}$ | $0.78_{\pm0.27}$ | $1.53_{\pm1.72}$ | $80.00_{\pm0.00}$ | $3.70_{\pm1.43}$ | $13.2_{\pm10.6}$ | $0.84_{\pm0.13}$ | $1.79_{\pm0.74}$ | $80.00_{\pm0.00}$ |
| SCORE | $1.20_{\pm3.46}$ | $4.2_{\pm10.7}$ | $0.95_{\pm0.14}$ | $0.43_{\pm1.06}$ | $35.30_{\pm0.95}$ | $5.60_{\pm3.92}$ | $21.2_{\pm16.1}$ | $0.78_{\pm0.14}$ | $2.25_{\pm0.78}$ | $35.80_{\pm0.98}$ |
| **DHT-CIT** | $\mathbf{0.00}_{\pm0.00}$ | $\mathbf{0.00}_{\pm0.00}$ | $\mathbf{1.00}_{\pm0.00}$ | $\mathbf{0.00}_{\pm0.00}$ | $\mathbf{9.00}_{\pm2.65}$ | $1.00_{\pm1.22}$ | $3.20_{\pm3.70}$ | $0.95_{\pm0.05}$ | $0.68_{\pm0.72}$ | $\mathbf{13.20}_{\pm4.30}$ |

| Method | Sin-20-20 Graph with Interventional Data ($\mathcal{D} = \{\boldsymbol{X}^0, \boldsymbol{X}^1\}$) | | | | | Sin-20-20 Graph with Observational Data ($\mathcal{D} = \{\boldsymbol{X}^1, \boldsymbol{X}^2\}$) | | | | |
|---|---|---|---|---|---|---|---|---|---|---|
| PC | $10.7_{\pm5.70}$ | $61.2_{\pm35.6}$ | $0.79_{\pm0.10}$ | $3.18_{\pm0.83}$ | - | $21.5_{\pm6.75}$ | $98.2_{\pm31.8}$ | $0.61_{\pm0.11}$ | $4.59_{\pm0.69}$ | - |
| FCI | $20.1_{\pm3.03}$ | $181._{\pm49.9}$ | $0.66_{\pm0.05}$ | $4.47_{\pm0.35}$ | - | $30.5_{\pm4.09}$ | $237._{\pm59.1}$ | $0.54_{\pm0.05}$ | $5.51_{\pm0.37}$ | - |
| GOLEM | $0.60_{\pm1.50}$ | $2.50_{\pm5.20}$ | $0.98_{\pm0.04}$ | $0.32_{\pm0.70}$ | - | $1.30_{\pm1.10}$ | $5.60_{\pm4.40}$ | $\mathbf{0.97}_{\pm0.03}$ | $0.93_{\pm0.66}$ | - |
| NOTEARS | $0.20_{\pm0.40}$ | $1.00_{\pm2.0}$ | $0.99_{\pm0.01}$ | $0.20_{\pm0.40}$ | - | $2.60_{\pm1.49}$ | $6.00_{\pm3.40}$ | $0.94_{\pm0.03}$ | $1.55_{\pm0.46}$ | - |
| ReScore | $0.90_{\pm2.70}$ | $3.90_{\pm11.7}$ | $0.98_{\pm0.06}$ | $0.30_{\pm0.90}$ | - | $2.00_{\pm0.77}$ | $5.10_{\pm2.90}$ | $0.95_{\pm0.01}$ | $1.38_{\pm0.28}$ | - |
| Granger | $103_{\pm17.2}$ | $371_{\pm8.19}$ | $0.10_{\pm0.01}$ | $10.1_{\pm0.89}$ | - | $104_{\pm20.7}$ | $368_{\pm8.82}$ | $0.10_{\pm0.03}$ | $10.1_{\pm1.01}$ | - |
| VarLiNGAM | $170_{\pm0.00}$ | $339_{\pm3.20}$ | $0.19_{\pm0.00}$ | $13.0_{\pm0.00}$ | - | $170_{\pm0.00}$ | $339_{\pm3.20}$ | $0.19_{\pm0.00}$ | $13.0_{\pm0.00}$ - | |
| CD-NOD | exceed 48h | - | - | - | - | - | - | - | - | - |
| CAM | $4.50_{\pm3.03}$ | $15.8_{\pm14.2}$ | $0.89_{\pm0.07}$ | $1.86_{\pm1.07}$ | $360.0_{\pm0.00}$ | $10.3_{\pm6.50}$ | $41.6_{\pm34.7}$ | $0.79_{\pm0.12}$ | $3.07_{\pm0.98}$ | $360.0_{\pm0.00}$ |
| SCORE | $0.20_{\pm0.63}$ | $0.90_{\pm2.70}$ | $0.99_{\pm0.02}$ | $0.14_{\pm0.45}$ | $170.1_{\pm0.32}$ | $7.40_{\pm2.41}$ | $31.3_{\pm21.7}$ | $0.85_{\pm0.04}$ | $2.68_{\pm0.47}$ | $172.1_{\pm0.22}$ |
| **DHT-CIT** | $\mathbf{0.00}_{\pm0.00}$ | $\mathbf{0.00}_{\pm0.00}$ | $\mathbf{1.00}_{\pm0.00}$ | $\mathbf{0.00}_{\pm0.00}$ | $\mathbf{16.44}_{\pm3.81}$ | $\mathbf{1.00}_{\pm1.32}$ | $\mathbf{3.10}_{\pm4.40}$ | $\mathbf{0.98}_{\pm0.03}$ | $\mathbf{0.51}_{\pm0.61}$ | $\mathbf{30.60}_{\pm7.70}$ |

* CD-NOD on **Sin-20-20** takes over 48 hours and **#Prune** on one-stage methods is not meaningful. We don't discuss these results and represent them with '-'.

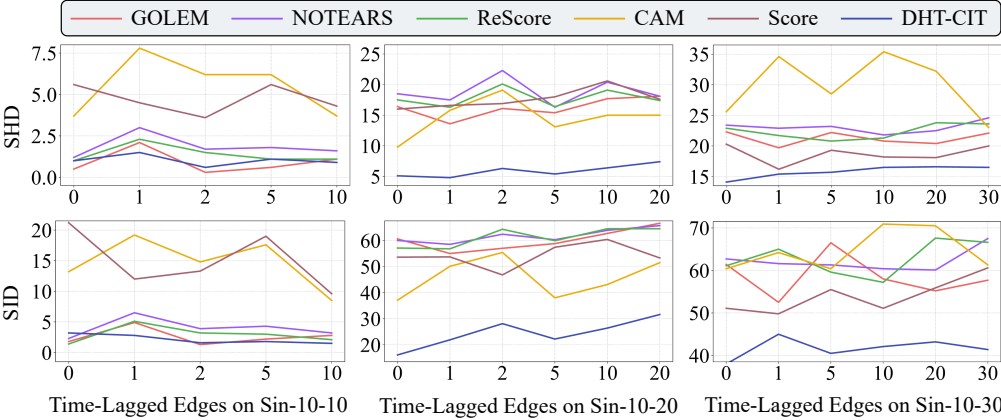

Figure 2: Exploring varying time-lagged edges and denser graph in observations $\mathcal{D} = \{\boldsymbol{X}^1, \boldsymbol{X}^2\}$.

dant hierarchical topology is much smaller than that of SCORE. On average, compared to SCORE, the number of pruned edges in DHT-CIT decreases 24.4 for **Sin-10-10** and 147.6 for **Sin-20-20**. As the underlying DAG's size increases, DHT-CIT achieves unbiased causal discovery on interventional data, but there may be a slight decrease on observational data, i.e., merely one error edge on average, but its F1-Score still exceeds 95%.

**Exploring Varying Time-Lagged Edges and Denser Graph in Observations** $\mathcal{D} = \{\boldsymbol{X}^1, \boldsymbol{X}^2\}$. In the experiments on denser graphs with more edges ($e = 2d$ and $e = 3d$), we gradually increase the number of time-lagged edges from other variables $\boldsymbol{X}_{-i}^{\tau-1}$ from 0 to $d$. As shown in Figure 3, most well-performed baselines on sparse graphs exhibit a substantial decrease in performance when applied to denser graphs. However, our DHT-CIT algorithm outperforms the best baseline on denser graphs. On the Sin-10-20 dataset, we achieve a 48% increase in SHD, a 48% increase in SID, and a 15% boost in F1-Score. On the Sin-10-30 dataset, we achieve a 30% increase in SHD, a 43% increase in SID, and a 7% boost in F1-Score. Our DHT-CIT is robust to varying time-lagged edges.

**Scaling to Different Noise Types**. To evaluate algorithm's robustness against various noise types, **Sin-10-10** data was generated with Laplace and Uniform noise. The results (Tab. 2) demonstrate the superior and robust performance of DHT-CIT against different types of noise, with the accuracy consistently comparable to that under Gaussian noise.

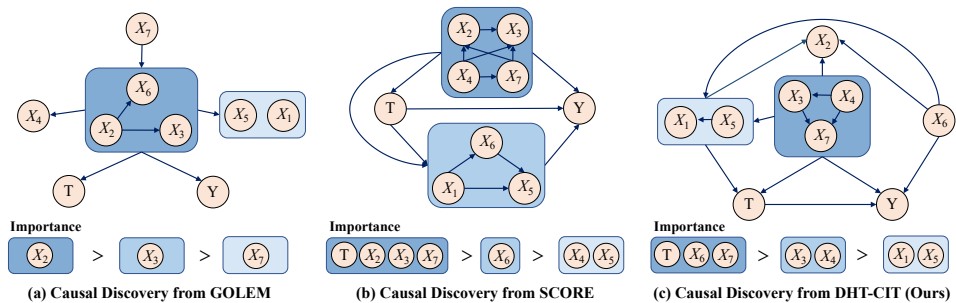

Figure 3: Causal Discovery on PM-CMR Dataset.

Table 2: The experiments on different noise type.

| | SHD↓ | SID↓ | F1-Score↑ | Dis.↓ | #Prune↓ |
|---|---|---|---|---|---|
| **Method** | \multicolumn{5}{c}{**Sin-10-10 data with Laplace noise ($\mathcal{D} = \{X^1, X^2\}$)**} |
| GOLEM | $1.50_{\pm1.20}$ | $\mathbf{2.80}_{\pm2.52}$ | $0.92_{\pm0.05}$ | $1.00_{\pm0.70}$ | - |
| NOTEARS | $1.60_{\pm0.06}$ | $3.70_{\pm3.10}$ | $0.92_{\pm0.03}$ | $1.23_{\pm0.26}$ | - |
| ReScore | $2.00_{\pm1.34}$ | $3.00_{\pm2.41}$ | $0.90_{\pm0.06}$ | $1.29_{\pm0.57}$ | - |
| CAM | $5.30_{\pm2.83}$ | $14.0_{\pm8.01}$ | $0.78_{\pm0.12}$ | $2.23_{\pm0.57}$ | $80.0_{\pm0.00}$ |
| SCORE | $3.90_{\pm1.70}$ | $9.90_{\pm6.01}$ | $0.84_{\pm0.06}$ | $1.93_{\pm0.43}$ | $35.5_{\pm0.92}$ |
| **DHT-CIT** | $\mathbf{1.20}_{\pm1.99}$ | $3.60_{\pm6.55}$ | $\mathbf{0.94}_{\pm0.04}$ | $\mathbf{0.59}_{\pm0.92}$ | $\mathbf{0.80}_{\pm1.40}$ |
| **Method** | \multicolumn{5}{c}{**Sin-10-10 data with Uniform noise ($\mathcal{D} = \{X^1, X^2\}$)**} |
| GOLEM | $2.60_{\pm1.80}$ | $6.80_{\pm3.94}$ | $0.89_{\pm0.06}$ | $1.46_{\pm0.68}$ | - |
| NOTEARS | $2.00_{\pm1.34}$ | $4.80_{\pm1.30}$ | $0.91_{\pm0.05}$ | $1.29_{\pm0.57}$ | - |
| ReScore | $1.70_{\pm0.90}$ | $3.70_{\pm2.90}$ | $0.92_{\pm0.04}$ | $1.21_{\pm0.48}$ | - |
| CAM | $8.90_{\pm7.15}$ | $21.4_{\pm12.0}$ | $0.68_{\pm0.22}$ | $2.14_{\pm0.73}$ | $80.0_{\pm0.00}$ |
| SCORE | $5.10_{\pm3.42}$ | $13.6_{\pm8.30}$ | $0.80_{\pm0.11}$ | $2.14_{\pm0.73}$ | $35.0_{\pm0.00}$ |
| **DHT-CIT** | $\mathbf{1.00}_{\pm2.19}$ | $\mathbf{1.10}_{\pm2.47}$ | $\mathbf{0.96}_{\pm0.09}$ | $\mathbf{0.44}_{\pm0.90}$ | $\mathbf{0.70}_{\pm1.55}$ |

Table 3: Average running time(s).

| | Sin-10-10 | Sin-10-20 | Sin-10-30 | Sin-20-20 |
|---|---|---|---|---|
| CD-NOD | 5433s | > 2h | > 2h | > 5h |
| CAM | 97.2s | 92.8s | 111.3s | 543.6s |
| GOLEM | 44.9s | 45.8s | 47.6s | 63.0s |
| SCORE | 40.7s | 38.1s | 45.2s | 193.1s |
| NOTEARS | 33.6s | 35.6s | 36.4s | 747.2s |
| ReScore | 24.1s | 23.2s | 24.2s | 29.2s |
| PC | 21.1s | 20.7s | 21.1s | 32.8s |
| FCI | 18.7s | 18.7s | 18.8s | 30.1s |
| VarLiNGAM | 17.9s | 18.3s | 18.4s | **27.3s** |
| Granger | **8.4s** | **8.5s** | 8.4s | 36.4s |
| **DHT-CIT** | 54.7s | 77.1s | 58.2s | 224.4s |

**Training Cost Analysis**. We implement 10 replications to study the average running time(s) of our DHT-CIT in a single execution in Tab. 3. Despite the increasing complexity and training cost, the DHT-CIT model maintains superior performance and scalability to larger and denser graphs, with a single execution time below 300 seconds, which is still within an acceptable threshold.

### 5.3 EXPERIMENTS ON REAL-WORLD DATA

The **PM-CMR** (Wyatt et al., 2020) is a public time series data that is commonly used to study the impact of the particle ($PM_{2.5}$, $T$) on the cardiovascular mortality rate (CMR, $Y$) in 2132 counties in the US from 1990 to 2010. Additionally, the dataset includes 7 variables ($X_{1:7}$) related to the city status, which are potential common causes of both $PM_{2.5}$ and CMR. The corresponding description of variables is detailed in Tab. 6 in Appendix E.3. With the prior knowledge, i.e., $T \leftarrow X_{1:7} \rightarrow Y$ and $T \rightarrow Y$, we draw two time-slices in 2000 & 2010 to evaluate the performance of the proposed DHT-CIT and two well-performed baselines (GOLEM and SCORE). As illustrated in Fig. 3, both GOLEM and SCORE do not generate true summary causal graph, and only our DHT-CIT achieves more accurate causal relationships in real-world data. GOLEM shows there is no direct edge from $T$ to $Y$ and SCORE shows that $T$ is the parent node of $\{X_1, X_5, X_6\}$, which contradicts the prior knowledge. Only our DHT-CIT algorithm recovers the dense causal graph, i.e., $T \leftarrow X_{1:7} \rightarrow Y$ and $T \rightarrow Y$. The results are consistent with the experiments on denser graphs: both GOLEM and SCORE are only applicable to sparse graphs, whereas our DHT-CIT maintains superior performance and scalability to larger and denser graphs. More detailed results are deferred to Appendix E.3.

## 6 CONCLUSION

In the subsampled time series with only two time-slices, conventional causal discovery methods designed for standard time series data would produce significant errors about the system's causal structure. To address this issue, we use some auxiliary instrumental variables in two time-slices as interventions to improve topological ordering and propose a novel DHT-CIT algorithm to learn a unique descendant hierarchical topology with merely a few spurious edges for identifying DAG of summary causal graph. The proposed DHT-CIT algorithm considerably eases the assumptions typically made in traditional time series studies about modeling causal structures at the system timescale, requiring causal sufficiency, and observing all time slices within the observation windows.

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

## A  RELATED WORK ON NON-TEMPORAL DATA

Constraint-based methods typically rely on conditional independence tests to identify causal relationships by testing the independence between variables given a set of conditions (Sun et al., 2007; Hyttinen et al., 2014), such as PC, FCI, SGS, and ICP (Spirtes et al., 2000; Zhang, 2008; Ramsey et al., 2012; Peters et al., 2016). Score-based methods (Tsamardinos et al., 2006; Ke et al., 2019; Zhu et al., 2020) search through the space of all possible causal structures with the aim of optimizing a specified metric, and rely on local heuristics to enforce the acyclicity, such as GES, and GIES (Chickering, 2002; Hauser & Bühlmann, 2012). Continuous-optimization methods (Zheng et al., 2018a; Lachapelle et al., 2020) view the search as a constrained optimization problem and apply first-order optimization methods to solve it, such as GraNDAG, GOLEM, NOTEARS, ReScore (Lachapelle et al., 2020; Ng et al., 2020; Zheng et al., 2018b; 2020; Zhang et al., 2023). Hybrid methods combine the advantages of both types of methods (Tsamardinos et al., 2006; Chen et al., 2021; Li et al., 2022; Hasan et al., 2023). GSP and IGSP algorithms (Solus et al., 2021; Wang et al., 2017) evaluate the score of each DAG structure using some information criterion and search for the optimal solution by iteratively changing permutations. Nevertheless, most constraint-based methods (e.g. PC, FCI) typically find causal structures within an equivalence class, resulting in a limited understanding of the underlying causal relationships. Score-based methods (e.g. NOTEARS, GES) rely on local heuristics to enforce acyclicity constraints, which can be insufficient for effectively handling large datasets. Additionally, the causal graphs produced by minimizing a specific score function are not guaranteed to be entirely accurate.

Recently, topology-based methods tackle the causal discovery problem by finding a certain topological ordering of the nodes and then pruning the spurious edges in topological ordering (Teyssier & Koller, 2005; Peters et al., 2014; Loh & Bühlmann, 2014; Park & Klabjan, 2017; Ghoshal & Honorio, 2018; Ahammad et al., 2021; Sanchez et al., 2022; Reisach et al., 2023). Examples of topology-based methods include CAM, SCORE and NoGAM (Bühlmann et al., 2014; Rolland et al., 2022; Montagna et al., 2023). These methods encounter a less combinatorial problem as the set of permutations is much smaller than the set of directed acyclic graphs. While these methods restrict the number and direction of potential edges in the learned DAG, they often generate numerous spurious edges that need to be pruned. In this paper, we focus on only two time-slices for learning causal relations and concatenate two time-slices data with the temporal edge to study a causal graph. Although the topology-based approach is widely applicable to cross-sectional data, its application to two time-slices studies is not routine and the data opportunities that two time-slices provide for topology-based methods are also overlooked.

## B  THE DIFFERENCE OF OUR THEOREMS WITH TRADITIONAL METHODS

**Our theorem differs from that of Peters et al. (2017):** Theorems 10.1, 10.2, 10.3, and 10.4 in Section 10 of Peters et al. (2017) rely on all previous time-slices within the observation windows that are accessible, i.e., $\mathbf{X}_{\text{past}(t)}$ could be observed, which are not satisfied in subsampled time-series. In Section 10.2.1, Peters et al. (2017) also noted that current methods for subsampled time-series require well-modeled interventions. Without well-defined interventions, it remains challenging to learn causal structures from subsampled time series without efficient solutions. In our Theorem 1, given two time-slice observations $\mathcal{D} = \{\boldsymbol{X}^{t_a}, \boldsymbol{X}^{t_b}\}_{t_a < t_b}$, $X_j^{t_b}$ is a descendant node of $X_i^{t_b}$ iff $X_i^{t_a} \not\perp X_j^{t_b} \mid \mathbf{an}_i^{t_a}$. We rigorously prove that it is both sound and complete under some graph constraints, i.e., acyclic summary causal graph, stationary full-time graph, and Markov property.

**Our theorem differs from that of Mastakouri et al. (2021):** Mastakouri et al. (2021) focuses on the detection of direct and indirect causes of a given target time series, rather than full graph discovery. Their theory requires that each observed candidate time series must be a non-descendant node of the target time series, and they need to conduct two conditional independence tests for each observed candidate to identify the direct causes of the target. In contrast, our theory not only constructs the Descendant Hierarchical Topology of the Directed Summary Causal Graph but also

requires only one conditional independence test per component. Furthermore, the full graph studied by Mastakouri et al. (2021) does not include instantaneous effects.

## C  THE DISCUSSION ABOUT THE COMPARED BASELINES

In this paper, we study the Directed Summary Causal Graph on Subsampled Time Series with instantaneous effects using only two time-slices, which is different from Standard Time Series Setting. Traditional methods designed for time series setting typically depend on modeling causal structures at the system timescale and assuming causal sufficiency, they require multiple time-slices with equal time intervals to estimate causal graphs. While Gong et al. (2015); Plis et al. (2015); Hyttinen et al. (2016) can identify a part of the causal information (i.e., an equivalence class) from subsampled time series data, they either rely on linear models or require the absence of instantaneous effects, which does not align with our setting. Most importantly, the algorithms designed for subsampled time series from these studies do not have reproducible open-source code available.

Since there are no available standard algorithms designed for two time-slice data, we develop variants of conventional non-temporal methods to the two time-slices setting proposed in this paper and compare the proposed algorithm with them: constraint-based methods, **PC** and **FCI** (Spirtes et al., 2000); score-based methods, **GOLEM** (Ng et al., 2020), **NOTEARS** with MLP (Zheng et al., 2020), and **ReScore** (Zhang et al., 2023); time-seires method, **CD-NOD** (Huang et al., 2020); topology-based methods, **CAM** (Bühlmann et al., 2014) and **SCORE** (Rolland et al., 2022).

**How to apply the non-temporal algorithms to the time series setting**: In the main experiments, we provide a broad range of time series variants of conventional non-temporal methods - namely PC, FCI, GOLEM, NOTEARS with MLP, ReScore + NOTEARS, CAM, and SCORE - that utilize a concatenation of the two cross-sectional data and the temporal edge (where the previous variable of the same components leads to the subsequent variable) as prior information (as initializing the adjacency matrix, and removing temporal edges that are not the same variable). Given $\mathcal{D} = \{\boldsymbol{X}^{t_a}, \boldsymbol{X}^{t_b}\}_{t_a < t_b}$, since the subsampled time series is a stationary stochastic process and the summary causal graph is acyclic, after removing the nodes $\boldsymbol{X}^{t_a}$ in the learned graph, the DAGs on $\boldsymbol{X}^{t_b}$ learned by the above causal discovery variants would approximate the summary causal graphs of interest.

Socre-based methods GOLEM, NOTEARS and ReScore are designed to recover the whole DAG by applying first-order optimization methods to solve a constrained optimization problem. While they may produce significant errors regarding the lagged effect, the causal graph learned on $\boldsymbol{X}^{t_b}$ is expected to approximate the true graph. Similarly, CAM relies on an additive structure to estimate a topological order by greedily maximizing data likelihood, and Score derives a topological order by approximating the score's Jacobian. Consequently, the causal graphs learned on $\boldsymbol{X}^{t_b}$ using these methods are also considered reliable in our setting. However, the time series variants of PC and FCI may struggle to identify causal graphs due to potential violations of causal sufficiency and time dependency. Although the identifiability results of these variants in two time-slices are not guaranteed, our experiments show that they significantly outperform traditional time series algorithms.

**Traditional temporal algorithms designed for time series setting**: Besides, we also provide a dynamic time series method CD-NOD (Huang et al., 2020), which applies the PC algorithm for causal discovery on an augmented dataset that includes a time label to capture unobserved changing factors. Furthermore, we also incorporate three traditional causal discovery methods designed for standard time series data: Granger causality (Granger, 1969; Shojaie & Michailidis, 2010), VARLiNGAM (Hyvärinen et al., 2010) as baselines in our main experiments[2].

## D  THE MOTIVATION AND PSEUDO-CODE OF OUR PROPOSED DHT-CIT

In many applications, the time series sampling process may be slower than the timescale of causal processes, resulting in numerous previous time-slices being missing or unreliable. In the presence of unmeasured time-slices, relying solely on a single time-slice is insufficient for identifying causal relations. Therefore, our motivation is to use just two reliable time-slices to explore the summary causal graph of subsampled time series, rather than depending on all previous time-slices that are

---

[2]For CD-NOD, Granger causality and VARLiNGAM, we use the latest implementation from the causal-learn package.

available and reliable (the limitation of traditional methods). In this paper, we demonstrate that if two valid time slices at two arbitrary moments are available, the variables in the earlier slice can be used as conditional instrumental variables to replace interventions and improve topological ordering. This method significantly relaxes the assumption inherent in traditional time series studies that depend on modeling causal structures at the system timescale, causal sufficiency, and all time slices in the observation windows could be observed (Granger, 1969; 1980; Luo et al., 2015; Nauta et al., 2019; Runge et al., 2019; Runge, 2020; Bussmann et al., 2021; Löwe et al., 2022; Assaad et al., 2022). However, it is important to note that if no previous time-slice data is available or reliable, our approach, like other causal discovery algorithms, will not produce identifiable results.

Under Assumptions 1, 2 and 3, we show how two time-slice help topological ordering for learning causal relations. In such cases, we propose DHT-CIT, a novel topological sorting algorithm that utilizes conditional independence tests per node to distinguish between its descendant and non-descendant nodes and build a unique descendant hierarchical topology with a few spurious edges for identifying summary causal graph. Algorithm 1 shows the pseudo-code of our DHT-CIT.

Hardware used: Ubuntu 16.04.3 LTS operating system with 2 * Intel Xeon E5-2660 v3 @ 2.60GHz CPU (40 CPU cores, 10 cores per physical CPU, 2 threads per core), 256 GB of RAM, and 4 * GeForce GTX TITAN X GPU with 12GB of VRAM.

Software used: Python 3.8 with cdt 0.6.0, ylearn 0.2.0, causal-learn 0.1.3, GPy 1.10.0, igraph 0.10.4, scikit-learn 1.2.2, networkx 2.8.5, pytorch 2.0.0.

---

**Algorithm 1** DHT-CIT: Descendant Hierarchical Topology with Conditional Independence Test

---

**Input:** Two time-slices $\mathcal{D} = \{\boldsymbol{X}^{t_a}, \boldsymbol{X}^{t_b}\}_{t_a < t_b}$ with $d$ nodes; two significance threshold $\alpha = 0.01$ and $\beta = 0.001$ for conditional independence test and pruning process; the layer index $k = 0$.
**Output:** One adjacency matrix of Descendant Hierarchical Topology $\boldsymbol{A}^{TP}$, one DAG $\mathcal{G}$.
**Components:** Conditional independence test $\mathbf{HSIC}(\dots)$; and pruning process $\mathbf{CAM}(\cdots)$.
**Stage 1 - Identifying Descendant Hierarchical Topology:**
**for** $i = 1$ **to** $d$ **do**
    Construct the conditional set $\boldsymbol{X}^{t_a}_{\otimes i}$ via an independence test $\boldsymbol{X}^{t_a}_{\otimes i} = \{X^{t_a}_j \mid X^{t_a}_j \perp X^{t_a}_i\}$
    **for** $j = 1$ **to** $d$ **do**
        $p_{i,j} = \mathbf{HSIC}(X^{t_a}_i, X^{t_b}_j \mid \boldsymbol{X}^{t_a}_{\otimes i})$
        $a^{TP}_{i,j} = \mathbb{I}(p_{i,j} \le \alpha)$
    **end for**
**end for**
We obtain $\boldsymbol{P} = \{p_{i,j}\}_{d \times d}$ and $\boldsymbol{A}^{TP} = \{a^{TP}_{i,j}\}_{d \times d}$
**Stage 2 - Adjusting the Topological Ordering:**
**while** The causal relationship between the unprocessed nodes is a directed cyclic graph **do**
    $k := k + 1$
    $X_{M_{i,k}} = \{X^{t_a}/X^{t_a}_i, \boldsymbol{L}_{1:k-1}\}$
    $X^{t_b}_i \in \boldsymbol{L}_k$, if $a^{TP}_{i,j} = 0$ for all $j \in M_{i,k}$
    **while** $\boldsymbol{L}_k = \emptyset$ **do**
        $p_{i^*,j^*} := 2\alpha$    and    $a^{TP}_{i^*,j^*} = 0$,    $(i^*, j^*) = \arg\max_{i,j}(p_{i,j} \le \alpha)$
        $X^{t_b}_i \in \boldsymbol{L}_k$, if $a^{TP}_{i,j} = 0$ for all $j \in M_{i,k}$
    **end while**
    We obtain $\boldsymbol{P} = \{p_{i,j}\}_{d \times d}$ and $\boldsymbol{A}^{TP} = \{a^{TP}_{i,j}\}_{d \times d}$
**end while**
**Stage 3 - Pruning Spurious Edges:**
We obtain $\mathcal{G} = \mathbf{CAM}(\mathcal{D}, \boldsymbol{A}^{TP}, \beta)$
**Return:** $\boldsymbol{A}^{TP}$ and $\mathcal{G}$

---

# E ADDITIONAL EXPERIMENTS ON SYNTHETIC AND REAL-WORLD DATASETS

## E.1 THE EXPERIMENTS ON MORE COMPLEX NON-LINEAR RELATIONSHIPS

**Datasets**. We test our algorithm on synthetic data generated from a *additive non-linear noise model* (Eq. 1) under Assumptions 1, 2 and 3. Given $d$ nodes and $e$ edges, we generate the causal graph $\mathcal{G}$

Table 4: The experiments on Sigmoid-10-10 & Poly-10-10 with observatios ( $\mathcal{D} = \{\boldsymbol{X}^1, \boldsymbol{X}^2\}$ )

| Method | Sigmoid-10-10 data with observational data ( $\mathcal{D} = \{\boldsymbol{X}^1, \boldsymbol{X}^2\}$ ) | | | | |
|---|---|---|---|---|---|
| | SHD↓ | SID↓ | F1-Score↑ | Dis.↓ | #Prune↓ |
| GOLEM | $4.30_{\pm 2.19}$ | $18.4_{\pm 7.92}$ | $0.78_{\pm 0.11}$ | $2.00_{\pm 0.51}$ | - |
| NOTEARS | $12.5_{\pm 5.40}$ | $45.3_{\pm 17.9}$ | $0.46_{\pm 0.21}$ | $3.44_{\pm 0.78}$ | - |
| ReScore | $12.2_{\pm 4.30}$ | $45.6_{\pm 14.4}$ | $0.45_{\pm 0.17}$ | $3.43_{\pm 0.63}$ | - |
| CAM | $3.70_{\pm 3.43}$ | $10.4_{\pm 7.86}$ | $0.82_{\pm 0.17}$ | $1.55_{\pm 1.20}$ | $80.00_{\pm 0.00}$ |
| SCORE | $9.90_{\pm 3.81}$ | $32.8_{\pm 11.6}$ | $0.56_{\pm 0.16}$ | $3.09_{\pm 0.61}$ | $38.90_{\pm 1.60}$ |
| **DHT-CIT** | $\mathbf{0.67}_{\pm 1.12}$ | $\mathbf{1.80}_{\pm 2.99}$ | $\mathbf{0.96}_{\pm 0.06}$ | $\mathbf{0.46}_{\pm 0.72}$ | $\mathbf{8.67}_{\pm 2.92}$ |
| Method | Poly-10-10 data with observational data ( $\mathcal{D} = \{\boldsymbol{X}^1, \boldsymbol{X}^2\}$ ) | | | | |
| GOLEM | $19.00_{\pm 4.00}$ | $59.4_{\pm 13.6}$ | $0.20_{\pm 0.12}$ | $4.33_{\pm 0.45}$ | - |
| NOTEARS | $17.8_{\pm 5.36}$ | $56.4_{\pm 16.9}$ | $0.23_{\pm 0.18}$ | $4.16_{\pm 0.64}$ | - |
| ReScore | $17.7_{\pm 4.73}$ | $57.3_{\pm 14.1}$ | $0.22_{\pm 0.15}$ | $4.16_{\pm 0.56}$ | - |
| CAM | $8.00_{\pm 4.69}$ | $19.8_{\pm 7.88}$ | $0.63_{\pm 0.21}$ | $2.68_{\pm 0.95}$ | $80.00_{\pm 0.00}$ |
| SCORE | $18.90_{\pm 4.33}$ | $40.4_{\pm 10.9}$ | $0.23_{\pm 0.13}$ | $4.32_{\pm 0.52}$ | $42.20_{\pm 1.48}$ |
| **DHT-CIT** | $\mathbf{3.22}_{\pm 3.15}$ | $\mathbf{10.8}_{\pm 5.69}$ | $\mathbf{0.84}_{\pm 0.15}$ | $\mathbf{1.51}_{\pm 1.03}$ | $\mathbf{11.33}_{\pm 3.87}$ |

using Erdos-Renyi model (Erdös & Rényi, 2011). In main experiments, we generate the data with Gaussian Noise for every variable $X_i^\tau$, $i = 1, 2, \cdots, d$ at time $\tau = 1, 2, \cdots, t$:

$$X_i^\tau = f_i \left(\mathbf{pa}_i^\tau, X_i^{\tau-1}, \mathbf{pa}_i^{\tau-1}\right) + \epsilon_i^\tau, \boldsymbol{\epsilon}^\tau \sim \mathcal{N}\left(0, 0.4 \cdot \mathrm{I}_d\right), \tag{6}$$

where $f_i$ is a twice continuously differentiable arbitrary function in each component, and $\mathrm{I}_d$ is a $d$ order identity matrix. To simulate real-world data as much as possible, we design 3 different non-linear functions non-linear($\cdot$) to discuss the performance of the HTS-CIT algorithm, for example:

$$\mathrm{Sin}(\mathbf{pa}_i^\tau) \quad = \quad \sum_{j \in \mathrm{pa}(X_i)} \sin(X_j^\tau), \tag{7}$$

$$\mathrm{Sigmoid}(\mathbf{pa}_i^\tau) \quad = \quad \sum_{j \in \mathrm{pa}(X_i)} \frac{3}{1 + \exp\left(-X_j^\tau\right)}, \tag{8}$$

$$\mathrm{Poly}(\mathbf{pa}_i^\tau) \quad = \quad \sum_{j \in \mathrm{pa}(X_i)} \frac{1}{10}\left(X_j^\tau + 2\right)^2. \tag{9}$$

In this paper, we use **Sin-$d$-$e$** to denote the synthetic dataset generated by non-linear function $\mathrm{Sin}(\cdot)$:

$$X_i^\tau = \mathrm{Sin}\left(\mathbf{pa}_i^\tau, X_i^{\tau-1}\right) + \frac{1}{10}\mathrm{Sin}\left(\mathbf{w} \cdot \mathbf{pa}_i^{\tau-1}\right) + \epsilon_i^\tau. \tag{10}$$

where $\mathrm{Sin}(\mathbf{pa}_i^\tau) = \sum_{j \in \mathrm{pa}(X_i)} \sin(X_j^\tau)$, $\mathrm{I}_d$ is a $d$-th order identity matrix, and $\mathbf{w}$ is a random 0-1 vector that controls the number and existence of time-lagged edges from $\mathbf{pa}_i^{\tau-1}$. Similarly, we define Sigmoid-$d$-$e$ and Poly-$d$-$e$.

**Results**. To simulate real-world data as much as possible, we design 2 additional non-linear functions to test the performance of our DHT-CIT, i.e., Sigmoid-$d$-$e$ & Poly-$d$-$e$. The results (Tab. 4) demonstrate that our DHT-CIT remains superior for other complex nonlinear functions with low error edges in identifying causal graphs. In addition, the number of spurious edges that must be pruned in the topological ordering is also minimal compared to CAM and SCORE.

### E.2 THE EXPERIMENTS ON LARGE GRAPHS WITH HIGH-DIMENSION VARIABLES

**Datasets**. Followed the data generation process (Eq. equation 5) in Section 5.2 in the main text. Given $d$ nodes and $e$ edges, we generate the causal graph $\mathcal{G}$ using the Erdos-Renyi model.

$$X_i^\tau = \mathrm{Sin}\left(\mathbf{pa}_i^\tau, X_i^{\tau-1}\right) + \frac{1}{10}\mathrm{Sin}\left(\mathbf{w} \cdot \mathbf{pa}_i^{\tau-1}\right) + \epsilon_i^\tau, \boldsymbol{X}^0 \sim \mathcal{N}\left(0, \mathrm{I}_d\right), \boldsymbol{\epsilon}^\tau \sim \mathcal{N}\left(0, 0.4 \cdot \mathrm{I}_d\right) \tag{11}$$

where $\mathrm{Sin}(\mathbf{pa}_i^\tau) = \sum_{j \in \mathrm{pa}(X_i)} \sin(X_j^\tau)$, $\mathrm{I}_d$ is a $d$-th order identity matrix, and $\mathbf{w}$ is a random 0-1 vector that controls the number and existence of time-lagged edges from $\mathbf{pa}_i^{\tau-1}$. In this experiment,

Table 5: The experiments on Sin-50-50 & Sin-100-100 datasets.

| Method | SHD↓ | SID↓ | F1-Score↑ | Dis.↓ | #Prune↓ | Running Time(s)↓ |
|---|---|---|---|---|---|---|
| | **Sin-50-50 data with Gauss noise ( $\mathcal{D} = \{X^1, X^2\}$ )** | | | | | |
| GOLEM | $87.9_{\pm 11.0}$ | $846.5_{\pm 166}$ | $0.24_{\pm 0.10}$ | $9.35_{\pm 0.58}$ | - | 1049.1s |
| ReScore | $83.5_{\pm 7.17}$ | $1044_{\pm 65.4}$ | $0.31_{\pm 0.06}$ | $9.13_{\pm 0.39}$ | - | 455.2s |
| SCORE | $17.4_{\pm 6.17}$ | $91.4_{\pm 49.7}$ | $0.85_{\pm 0.04}$ | $4.11_{\pm 0.74}$ | $1175_{\pm 0.32}$ | **143.3s** |
| **DHT-CIT(50% Intervention)** | $21.3_{\pm 6.85}$ | $102._{\pm 64.3}$ | $0.81_{\pm 0.05}$ | $4.56_{\pm 0.64}$ | $175._{\pm 8.17}$ | 829.1s |
| **DHT-CIT(80% Intervention)** | $18.5_{\pm 5.62}$ | $97.2_{\pm 39.0}$ | $0.84_{\pm 0.05}$ | $4.11_{\pm 0.57}$ | $86.6_{\pm 7.52}$ | 625.7s |
| **DHT-CIT(100% Intervention)** | $\mathbf{16.4}_{\pm 4.40}$ | $\mathbf{88.4}_{\pm 37.1}$ | $\mathbf{0.86}_{\pm 0.04}$ | $\mathbf{4.00}_{\pm 0.58}$ | $\mathbf{58.8}_{\pm 8.52}$ | 327.1s |
| Method | **Sin-100-100 data with Gauss noise ( $\mathcal{D} = \{X^1, X^2\}$ )** | | | | | |
| GOLEM | $160.6_{\pm 17.2}$ | $1898_{\pm 764.1}$ | $0.25_{\pm 0.08}$ | $12.6_{\pm 0.68}$ | - | 3904.2s |
| ReScore | $163.3_{\pm 13.7}$ | $4009_{\pm 549.3}$ | $0.34_{\pm 0.05}$ | $12.7_{\pm 0.55}$ | - | 578.2s |
| SCORE | $32.6_{\pm 4.71}$ | $211.4_{\pm 39.7}$ | $0.86_{\pm 0.02}$ | $5.70_{\pm 0.42}$ | $4450_{\pm 0.47}$ | **149.2s** |
| **DHT-CIT(50% Intervention)** | $37.6_{\pm 6.30}$ | $219.4_{\pm 64.3}$ | $0.83_{\pm 0.02}$ | $6.11_{\pm 0.49}$ | $306.4_{\pm 17.6}$ | 3204.2s |
| **DHT-CIT(80% Intervention)** | $29.5_{\pm 6.26}$ | $180.9_{\pm 47.2}$ | $0.87_{\pm 0.02}$ | $5.40_{\pm 0.56}$ | $248.0_{\pm 27.0}$ | 1187.2s |
| **DHT-CIT(100% Intervention)** | $\mathbf{26.7}_{\pm 6.80}$ | $\mathbf{160.5}_{\pm 55.3}$ | $\mathbf{0.88}_{\pm 0.03}$ | $\mathbf{5.13}_{\pm 0.63}$ | $\mathbf{197.1}_{\pm 24.8}$ | 761.2s |

we set the number of time-lagged edges from other variables as 0. To evaluate our DHT-CIT on a larger graph with 50/100 nodes, we generate large graph **Sin-50-50** and **Sin-100-100**.

Although theoretically, DHT-CIT can achieve unbiased estimation, it is limited by the performance of conditional independence tests. For the conditional instrumental variables described above, we calculate the conditional independencies using the conditional independence HSIC test from (Zhang et al., 2011) with Gaussian kernel. However, as the data dimension increases, the accuracy of the HSIC test decreases, leading to incorrect topological orderings generated by DHT-CIT. To mitigate this issue, given two-time slices ( $\mathcal{D} = \{X^1, X^2\}$ ), we implement random intervention to some nodes in the previous states of two time-slices, and then apply DHT-CIT to identify potential directed acyclic graphs. Based on the percentage of intervened nodes in the previous state, we refer to it as **DHT-CIT(50% Intervention)**, **DHT-CIT(80% Intervention)**, and **DHT-CIT(100% Intervention)**. Note that the time consumption of NOTEARS-MLP and CAM increases substantially (exceeds 5000s) as the graph size increases, thus, we do not implement NOTEARS-MLP and CAM.

**Results**. From the results on larger graphs (**Sin-50-50** and **Sin-100-100**) in Tab. 5, we have the following observation: (1) GOLEM and ReScore fail to identify the true DAG on larger graphs; (2) In terms of pruning efficiency, DHT-CIT outperforms SCORE by providing a more effective hierarchical topological ordering. The number of edges to be pruned in the topological ordering learned by SCORE is at least 10 times greater that of the proposed DHT-CIT, which greatly increases the workload for subsequent pruning processes; (3) With at least 50% intervened nodes in the previous stats, DHT-CIT(50% Intervention) can produce results that are comparable to the most advanced methods SCORE. As the proportion of intervened nodes in the previous stats increases (exceed 80%), our approach (DHT-CIT(80% Intervention) and DHT-CIT(100% Intervention)) will gradually outperform SCORE. Two time-slices with random intervention will help to improve the identification of the topological ordering of the underlying DAG.

### E.3 THE EXPERIMENTS ON REAL-WORLD DATASET

The **PM-CMR**[3] (Wyatt et al., 2020) is a public time series data that is commonly used to study the impact of the particle ($PM_{2.5}$, $T$) on the cardiovascular mortality rate (CMR, $Y$) in 2132 counties in the US from 1990 to 2010. Additionally, the dataset includes 7 variables ($X_{1:7}$) related to the city status, which are potential common causes of both $PM_{2.5}$ and CMR. The corresponding description of variables is detailed in Tab. 6. With the prior knowledge, i.e., $T \leftarrow X_{1:7} \rightarrow Y$ and $T \rightarrow Y$, we draw two time-slices in 2000 & 2010 to evaluate the performance of the proposed DHT-CIT and two well-performed baselines (GOLEM and SCORE).

**Results**. With the prior knowledge, i.e., $T \leftarrow X_{1:7} \rightarrow Y$ and $T \rightarrow Y$, we draw two time-slices in 2000 & 2010 to evaluate the performance of the proposed DHT-CIT and two well-performed baselines (GOLEM and SCORE). As illustrated in Fig. 3 and 4, both GOLEM and SCORE do

---

[3]PM-CMR:https://pasteur.epa.gov/uploads/10.23719/1506014/SES_PM25_CMR_data.zip

Table 6: The Description for Real Variables on PM-CMR Dataset.

| Variable | Description |
|---|---|
| $\text{PM}_{2.5}(T)$ | Annual county PM2.5 concentration, $\mu g/m^3$ |
| $\text{CMR}(Y)$ | Annual county cardiovascular mortality rate, deaths/100,000 person-years |
| $\text{Unemploy}(X_1)$ | Civilian labor force unemployment rate in 2010 |
| $\text{Income}(X_2)$ | Median household income in 2009 |
| $\text{Female}(X_3)$ | Family households - female householder, no spouse present in 2010 / Family households in 2010 |
| $\text{Vacant}(X_4)$ | Vacant housing units in 2010 / Total housing units in 2010 |
| $\text{Owner}(X_5)$ | Owner-occupied housing units - percent of total occupied housing units in 2010 |
| $\text{Edu}(X_6)$ | Educational attainment - persons 25 years and over - high school graduate (includes equivalency) in 2010 |
| $\text{Poverty}(X_7)$ | Families below poverty level in 2009 |

**The State in 2010 (GOLEM)** — The State in 2000

|  | T | Y | X1 | X2 | X3 | X4 | X5 | X6 | X7 |
|---|---|---|---|---|---|---|---|---|---|
| T | 0 | 0 | 0 | 0 | 0 | 0 | 0 | 0 | 0 |
| Y | 0 | 0 | 0 | 0 | 0 | 0 | 0 | 0 | 0 |
| X1 | 0 | 0 | 0 | 0 | 0 | 0 | 0 | 0 | 0 |
| X2 | 1 | 1 | 0 | 0 | 1 | 1 | 0 | 1 | 0 |
| X3 | 1 | 1 | 1 | 0 | 0 | 0 | 1 | 0 | 0 |
| X4 | 0 | 0 | 0 | 0 | 0 | 0 | 0 | 0 | 0 |
| X5 | 0 | 0 | 0 | 0 | 0 | 0 | 0 | 0 | 0 |
| X6 | 0 | 0 | 0 | 0 | 0 | 0 | 1 | 0 | 0 |
| X7 | 0 | 0 | 0 | 1 | 1 | 0 | 0 | 1 | 0 |

**The State in 2010 (SCORE)**

|  | T | Y | X1 | X2 | X3 | X4 | X5 | X6 | X7 |
|---|---|---|---|---|---|---|---|---|---|
| T | 0 | 1 | 1 | 0 | 0 | 0 | 1 | 1 | 0 |
| Y | 0 | 0 | 0 | 0 | 0 | 0 | 0 | 0 | 0 |
| X1 | 0 | 0 | 0 | 0 | 0 | 0 | 1 | 1 | 0 |
| X2 | 1 | 1 | 1 | 0 | 1 | 0 | 1 | 1 | 0 |
| X3 | 1 | 1 | 1 | 0 | 0 | 0 | 1 | 1 | 0 |
| X4 | 1 | 0 | 1 | 1 | 1 | 0 | 1 | 1 | 1 |
| X5 | 0 | 0 | 0 | 0 | 0 | 0 | 0 | 0 | 0 |
| X6 | 0 | 1 | 0 | 0 | 0 | 0 | 1 | 0 | 0 |
| X7 | 1 | 1 | 1 | 1 | 1 | 0 | 1 | 1 | 0 |

**The State in 2010 (DHT-CIT)**

|  | T | Y | X1 | X2 | X3 | X4 | X5 | X6 | X7 |
|---|---|---|---|---|---|---|---|---|---|
| T | 0 | 1 | 0 | 0 | 0 | 0 | 0 | 0 | 0 |
| Y | 0 | 0 | 0 | 0 | 0 | 0 | 0 | 0 | 0 |
| X1 | 1 | 0 | 0 | 0 | 0 | 0 | 0 | 0 | 0 |
| X2 | 0 | 0 | 0 | 0 | 0 | 0 | 0 | 0 | 0 |
| X3 | 1 | 0 | 1 | 1 | 0 | 0 | 1 | 0 | 1 |
| X4 | 1 | 0 | 0 | 1 | 1 | 0 | 1 | 0 | 1 |
| X5 | 0 | 0 | 1 | 0 | 0 | 0 | 0 | 0 | 0 |
| X6 | 0 | 1 | 1 | 1 | 0 | 0 | 1 | 0 | 0 |
| X7 | 1 | 1 | 0 | 0 | 0 | 0 | 0 | 0 | 0 |

Figure 4: Learned Adjacency Matrix on PM-CMR Dataset.

not generate true summary causal graphs, and only our DHT-CIT achieves more accurate causal relationships in real-world data. GOLEM shows there is no direct edge from $T$ to $Y$ and SCORE shows that $T$ is the parent node of $\{X_1, X_5, X_6\}$, which contradicts the prior knowledge. Only our DHT-CIT algorithm recovers the dense causal graph, i.e., $T \leftarrow X_{1:7} \rightarrow Y$ and $T \rightarrow Y$. The results are consistent with the experiments on denser graphs: both GOLEM and SCORE are only applicable to sparse graphs, whereas our DHT-CIT maintains superior performance and scalability to larger and denser graphs.

Notably, our algorithm demonstrates significant improvement on denser graphs (Fig. 2). In comparison to the best baseline, our algorithm boasts a 48% increase in SHD, a 48% increase in SID, and a 15% boost in F1-Score on **Sin-10-20**, and boasts a 30% increase in SHD, a 43% increase in SID, and a 7% boost in F1-Score on **Sin-10-30**. Most previous baselines were only applicable to sparse graphs, whereas our algorithm exhibits substantial improvements on dense graphs. Therefore, we believe that DHT-CIT provides a more precise DAG for the PM-CMR dataset. Therefore, to effectively combat cardiovascular disease, it is recommended that cities disseminate information about its dangers, promote prevention, and provide medical care for low-income families.

# F  RELAXING MARKOV ASSUMPTION TO HIGH-ORDER MARKOV MODELS

Notably, in this paper, we can relax the Markov Assumption to a high-order Markov Assumption. This means that the future time-slice $X^{t+1}$ depends only on states $X^{t \cdots t-q+1}$ and does not directly depend on states $X^{1 \cdots t-q}$. Then, with $q+1$ time-slices ($X^{t_a \cdots t_a-q+1}$ and $X^{t_b}$), we can use $\mathbf{an}^{t_a \cdots t_a-q+1}$ to replace the condition set $\mathbf{an}^{t_a}$ to infer the Descendant-oriented Conditional Independence Criteria (Theorem 1). However, in this paper, we focus exclusively on the two-time-slices algorithm to demonstrate our theorem and algorithm. We have included a section to discuss the extension of the Markov Assumption to a high-order Markov Assumption in the Appendix.

# G  THE ADVANTAGES AND LIMITATIONS OF USING TWO TIME-SLICES DATA

Advantage: (1) Enhanced Topological Ordering: Traditional topology-based methods typically produce non-unique topological orderings with numerous spurious edges, resulting in decreased accuracy and efficiency in downstream search tasks. By using two time-slices as auxiliary instrumental variables, we can learn causal relations more efficiently, with a reduced search space and fewer spurious edges. (2) Feasibility in Intervention-Limited Contexts: Using interventional data can quickly identify (non-)descendants for each node and construct a more precise topological ordering. In scenarios where interventions are infeasible, unethical, or too costly, using two time-slices to replace intervention can be a practical alternative. (3) Reduced Data Requirements: In time series scenarios, traditional methods depend on the modeling causal structures at the system timescale, causal sufficiency, and all time slices in the observation windows could be observed. In this paper, we propose exploring limited time-slices, i.e., two reliable time-slices, to ease the data requirements. (4) Scaling to Non-linear and Non-Gaussian Models: We use two time-slices as conditional instrumental variables to simulate exogenous interventions. When applied to a variable, these simulated interventions only affect the variable's value, and the permutation would propagate to its descendant nodes. Then, we can apply conditional independence tests to capture these intervention-related permutations for identifying each variable's descendants, without requiring any structural or distributional assumptions about the data.

Limitations: Our DHT-CIT algorithm strictly relies on the Acyclic Summary Causal Graph and Consistency Throughout Time Assumptions, which are common in time series studies [Assaad et al., 2022]. Additionally, if no previous time-slice data is available or reliable, both the two time-slices approach and other causal discovery algorithms will not yield identifiable results.

# H  FUTURE APPLICATIONS

## H.1  POTENTIAL APPLICATIONS OF OUR DHT-CIT ALGORITHM

The proposed DHT-CIT can be integrated as a module into any existing topology-based method to enhance the topological ordering. Additionally, our DHT-CIT algorithm is capable of identifying the true causal graph from the Markov equivalence classes that are typically learned using traditional methods. For instance, our DHT-CIT algorithm can effectively use the Descendant Hierarchical Topology to orient the undirected edges outputted by a constraint-based algorithm such as PCMCI (Runge et al., 2019; Runge, 2020).

## H.2  GENERALIZING THE DHT-CIT ALGORITHM TO OTHER DOMAINS

As long as the common time series causal assumptions in (Assaad et al., 2022) and the q-order Markov Assumption are satisfied, we can directly extend our algorithm to other domains and applications with q+1 time-slices ($X^{t_a \cdots t_a - q + 1}$ and $X^{t_b}$). For example, we can analyze the causal graph of city status variables in PM-CMR (Wyatt et al., 2020), and explore the relationships between various factors affecting soil moisture. In human genomics and gene expression, we also can establish two-time-slices causal relationships (surjections: where each expression variable can find a corresponding conditional instrumental variable in the genomic sequence variables). The challenges arise as different time series data may adhere to various high-order Markov Assumptions, which we need to identify. Additionally, sometimes the temporal transfer of events/processes might conceal causal relationships, requiring further extraction, such as the two-time-slices causal relationships between genomic and gene expression data. The DHT-CIT algorithm provides an excellent tool and opportunity for identifying the topological ordering in the aforementioned forms of data. We haven't covered experiments in all areas in this paper due to the high cost of data acquisition. Future updates on these datasets will be shared on our project pages.

