# OpenReview forum: "Two Time-Slices Help Topological Ordering for Learning Directed Acyclic Graphs"
_ICLR.cc/2024/Conference — Submitted to ICLR 2024_

### Official Review · Reviewer_mehL · 2023-10-28

**Soundness:** 2 fair
**Presentation:** 2 fair
**Contribution:** 2 fair
**Rating:** 5
**Confidence:** 3

**Summary:**

The paper addresses the problem of causal discovery in a time series setting where data from two time slices are given. To do this, the authors propose an approach that identifies the topological order and results in a smaller number of potential edges, which simplifies the pruning problem. The suggested approach utilizes independence tests and the lagged time series structure. In the experiments, the authors compare different causal discovery approaches on artificial data and one real-world dataset.

**Strengths:**

- Provides a broad overview of different causal discovery works
- Figures are helpful for illustrating some of the concepts
- The authors make some good efforts for a broad comparison in the experiments, although this could be improved (see Question section)

**Weaknesses:**

- Unclear positioning of the paper. It addresses causal discovery with time series data but has very limited discussion of related work in the time series domain. Section 10 of the book "Elements of Causal Inference" could help in formulating a clearer problem statement.
- Novelty is unclear, since Theorem 1 seems to follow directly from the assumed lagged time structure (see Question section for details).
- While there is comparison with a broad range of methods, they are mostly designed for the IID setting rather than the time series setting.
- The functional form in Eq. (1) is more restrictive than the referenced additive noise model class.

**Questions:**

My main concern is the unclear positioning of the paper. It refers to DAGs and compares approaches designed for IID data (e.g. PC, FCI, etc.) but in a time series setting, mixing the definitions. A much cleaner definition of the problem setting and introduction of "time slices" is needed, especially the connection between observations at different times as interventions.

Some remarks and questions:
- You refer to "unique orderings" of the topology. However, this appears impossible since equivalent orderings exist (e.g. X → Y ← Z has completely equivalent orderings (X,Z,Y) and (Z,X,Y)). If the "uniqueness" refers to something else, then this should be clarified early on.
- After the introduction, it is unclear why the problem isn't an edge pruning problem, as mentioned later. It is unclear why the 'naive' approach of pruning edges after obtaining the causal order is insufficient.
- Consider avoiding the use of notations that has not yet been introduced, such as X^t-3 in the introduction.
- You write "two time-slides data" at the end of the introduction. What is a "slide" here?
- The related work is mostly IID approaches with very limited discussion of time series approaches. Why is this?
- You mention causal discovery algorithms identify the equivalence class, which is true for some (e.g. PC, FCI) but not all (e.g. NOTEARS aims to recover the whole DAG).
- When you mention 'DAG', do you refer to the summary graph? This should be clarified.
- Defining X_i as a random variable does not make sense in the time series setting, where X_i^t is a random variable (with typically only one observation). See Section 10 of "Elements of Causal Inference".
- Equation (1) is more restrictive than an additive noise model. An additive noise model is defined as Y = f(instantaneous parents, all other lagged parents) + N in a time series setting. In your case, you restrict the lagged variables to be separate functions and has only an additive influence. This seems quite restrictive.
- The time slice definition is confusing. You mention an initial slice from 1 to t-1 and the present is t. Where is the gap between them?
- Assumption 1 seems flawed. X^t should be independent of X^t-i-k given X^t-1, ..., X^t-i since any previous lag could directly impact the current time lag (skip connection), rendering them dependent.
- Definition 2 is unclear. What about X → Y → Z? It has 3 layers (X)_L1, (Y)_L2, (Z)_L3. By the definition, L1 > L3, so X has a direct edge to Z, which would be wrong?
- Theorem 1 seems to follow directly from the lagged structure and Markov property. A discussion comparing it to Section 10.3 of "Elements of Causal Inference" and the paper "Necessary and sufficient conditions for causal feature selection in time series with latent common causes" by Mastakouri et al. would be helpful.
- Unclear why a time slice is an intervention. It seems a time slice just means recorded at different times (i.e. with a time gap)?
- How do you apply the IID algorithms like PC and FCI to the time series setting? Do you just treat it as IID?
- While the experiments have many approaches in the comparison and different data set sizes, more relevant comparisons to time series methods like Granger causality or VARLiNGAM would be insightful. The functional relationships could also be more general, like neural networks with random weights to generate arbitrary non-linear connections.

---

> ### Author Response · Authors · 2023-11-17
> **Responses by Authors (Part 1): About the Main Concern - Definition of the problem setting**
>
> Dear reviewer,
>
> Thank you for your insightful feedback and valuable suggestions, which have significantly enhanced our manuscript, especially in defining the problem setting, positioning this paper, discussing baselines, and clarifying the novelty and contributions of our theorem. We have posted point-to-point replies to each question/comment raised by you and uploaded the revised version of our paper (with track changes marked in blue). We sincerely hope that we could address your concerns. We look forward to any additional comments you might have that could further refine our paper.
>
> > **[Q1] A much cleaner definition of the problem setting and introduction of "time slices" is needed, especially the connection between observations at different times as interventions.**
>
> **Response:** This paper studies the **summary causal graph on subsampled time series with instantaneous effects using only two time-slices**, in which time-slices (cross-sectional observations) are sampled at a coarser timescale than the causal timescale of the underlying system. In this paper, **a time-slice denotes sampled cross-sectional observations on multiple samples** captured at a specific time point or time instant (within the given measurement accuracy) on the timeline of some system.
>
> **Subsampled Time Series with Only Two Time-slices** (Figure 1(b)) is a more challenging task than the **Standard Subsampled Time Series problem** (Figure 10.4 in Section 10.2.1 of "Elements of Causal Inference"). **The challenges arise from**: (1) We only observe two time-slices from the Subsampled Time Series, and numerous unmeasured time slices may be latent in the time series; (2) We focus on Subsampled Time Series with instantaneous effects, thus, the theorems of Granger causality, "Elements of Causal Inference" and "Necessary and sufficient conditions for causal feature selection in time series with latent common causes" are not applicable [Peters et al., 2017; Mastakouri et al., 2021]; (3) We cannot rely on full interventions to identify the Directed Summary Causal Graph, as full interventions are often expensive, unethical, or even infeasible. Therefore, **we explore using two time-slices as a substitute for intervention data to improve causal ordering and construct a unique Descendant Hierarchical Topology**.
>
> **The connection between observations at different times as interventions:** As indicated by red marking in Figure 1(b), an intervention on the variable $X_2^{t-3}$ would transfer its effect to descendants ($X_3^{t-3}, X_4^{t-3}$), but not to non-descendants ($X_1^{t-3}, X_5^{t-3}$). This helps in swiftly identifying each variable's descendants, enhancing topological ordering. Inspired by this concept, as illustrated in Fig. 1(b), where each node is influenced by its ancestors and itself, each variable in the earlier time-slice would transmit self-perturbations to both itself and its descendants in the subsequent time-slice, serving as simulated interventions.
>
> **We have revised and rephrased the connection between our work and existing methods in the Introduction and Related Work sections, highlighted in blue, to further clarify our paper's position in the field.**

---

> > ### Author Response · Authors · 2023-11-17
> > **Responses by Authors (Part 2): About the Main Concern - Unclear positioning of the paper.**
> >
> > > **[Q2 & W1] Unclear positioning of the paper. It addresses causal discovery with time series data but has very limited discussion of related work in the time series domain. Section 10 of the book "Elements of Causal Inference" could help in formulating a clearer problem statement.**
> >
> > **Response:** Thanks for your advice. In this paper, we study the **Directed Summary Causal Graph on Subsampled Time Series with instantaneous effects using only two time-slices**, which is different from Standard causal discovery methods for estimating causal structure from Subsampled Time Series. We have revised the discussion of related work about subsampled time series in the Introduction and Related Work Sections:
> >
> > Standard methods for inferring causal structure from conventional time series typically **focus either on estimating a transition model at the measurement timescale** (e.g., Granger causality [Granger, 1969; Granger, 1980]) or they **integrate a model of measurement timescale with 'instantaneous' or 'contemporaneous' causal relations** to capture interactions within and between d-variate time series from observational data [Lutkepohl, 2005; Hyvarinen et al., 2010; Luo et al., 2015; Nauta et al., 2019; Runge et al., 2019; Runge et al., 2020; Bussmann et al., 2021; Lowe et al., 2022; Assaad et al., 2022]. However, these methods depend on modeling causal structures at the system timescale and assuming causal sufficiency. Both of these conditions might not hold in Subsampled Time Series with only two time-slices, as there could be numerous unmeasured time slices latent in the time series, either before or between these two observed time-slices [Section 10.2.1, Peters et al., 2017].
> >
> > **Subsampled processes with a few time-slices in time series setting are ubiquitous and inherent in the real world, however, causal discovery over Subsampled Time Series is not as well explored.** With the prior of the degree of undersampling, [Gong et al., 2015] uses Expectation-Maximization algorithm to recover the linear temporal causal relations from the subsampled data. [Tank et al., 2019] take structural vector autoregressive models for parameter identifiability and estimation. The identifiability of both works is achieved only for linear data. As for nonlinear data, [Danks and Plis, 2013] only can extract constraints on summary graphs from the strongly connected components (SCC). Inspired by [Gong et al., 2015; Plis and Danks, 2015], [Hyttinen et al., 2016] proposes a constraint optimization approach to identify a part of the causal information (i.e., an equivalence class) from subsampled time series data, but requires no instantaneous effects.
> >
> > **We have divided the Related Work Section into two parts: one focusing on temporal data is included in the main text, while the other, dealing with non-temporal data, is positioned in Appendix A.**

---

> > > ### Author Response · Authors · 2023-11-17
> > > **Responses by Authors (Part 3): About the Main Concern - How to apply the non-temporal algorithms to two time-slices setting:**
> > >
> > > > **[Q3 & W3] While there is comparison with a broad range of methods, they are mostly designed for the IID setting rather than the time series setting.**
> > >
> > > **Response - There are no available standard algorithms designed for two time-slice data**: In this paper, we study the Directed Summary Causal Graph on Subsampled Time Series with instantaneous effects using only two time-slices, which is different from Standard Time Series Setting. Traditional methods designed for time series settings typically depend on modeling causal structures at the system timescale and assuming causal sufficiency, they require multiple time-slices with equal time intervals to estimate causal graphs. While [Gong et al., 2015; Plis and Danks, 2015; Hyttinen et al., 2016] can identify a part of the causal information (i.e., an equivalence class) from subsampled time series data,  they either rely on linear models or require the absence of instantaneous effects, which does not align with our setting. Most importantly, the algorithms designed for subsampled time series from these studies do not have reproducible open-source code available.
> > >
> > > Notably, most algorithms designed for causal discovery in time series data are variants of conventional non-temporal methods. **Since there are no available standard algorithms designed for two time-slice data, we develop variants of conventional non-temporal methods** to the two time-slices setting proposed in this paper and compare the proposed algorithm with them.
> > >
> > > > **[Q4 & W3] How do you apply the IID algorithms like PC and FCI to the time series setting? Do you just treat it as IID? It refers to DAGs and compares approaches designed for IID data (e.g. PC, FCI, etc.) but in a time series setting, mixing the definitions.**
> > >
> > > **Response:** Thank you for your question, **we have included an additional section to discuss them in Appendix C**, and the related discussion also could be found in the related work, experiment, and result Sections.
> > >
> > > **How to apply the non-temporal algorithms to the time series setting:** In the main experiments without time-lagged edges from $\mathbf{pa}_{i}^{\tau-1}$, i.e.,  $\mathbf{w}=\mathbf{0}$, we provide **a broad range of time series variants of conventional non-temporal methods** -  namely PC, FCI, GOLEM, NOTEARS with MLP, ReScore + NOTEARS, CAM, and SCORE - that **utilize a concatenation of the two cross-sectional data and the temporal edge** (where the previous variable of the same components leads to the subsequent variable) as prior information (as initializing the adjacency matrix, and removing temporal edges that are not the same variable). Given the following two time-slices data
> > >
> > > $\mathcal{D} = \\{ \boldsymbol{X}^{t_a}, \boldsymbol{X}^{t_b} \\}_{t_a < t_b}$,
> > >
> > > since the subsampled time series is a stationary stochastic process and the summary causal graph is acyclic, after removing the nodes $\boldsymbol{X}^{t_a}$, the DAGs on $\boldsymbol{X}^{t_b}$ learned by the above causal discovery variants just are the summary causal graphs of interest.
> > >
> > > Socre-based methods GOLEM, NOTEARS and ReScore are designed to recover the whole DAG by applying first-order optimization methods to solve a constrained optimization problem. While they may produce significant errors regarding the lagged effect, the causal graph learned on $\boldsymbol{X}^{t_b}$ is expected to approximate the true graph. Similarly, CAM relies on an additive structure to estimate a topological order by greedily maximizing data likelihood, and Score derives a topological order by approximating the score’s Jacobian. Consequently, the causal graphs learned on $\boldsymbol{X}^{t_b}$ using these methods are also considered reliable in our setting. However, the time series variants of PC and FCI may struggle to identify causal graphs due to potential violations of causal sufficiency and time dependency. **Although the identifiability results of these variants in two time-slices are not guaranteed, our experiments show that they significantly outperform traditional time series algorithms.**
> > >
> > > **Besides, we also provide a dynamic time series method CD-NOD**, which applies the PC algorithm for causal discovery on an augmented dataset that includes a time label to capture unobserved changing factors. According to your advice, we also incorporate **Granger causality [Granger et al., 1969; Shojaie et al., 2010] and VARLiNGAM [Hyvärinen et al., 2010] as baselines** in our main experiments.

---

> ### Author Response · Authors · 2023-11-17
> **Responses by Authors (Part 4): About the Novelty, Theorem, and Data Generation Function**
>
> > **[Q5 & W2] Novelty is unclear. Theorem 1 seems to follow directly from the lagged structure and Markov property. A discussion comparing it to Section 10.3 of "Elements of Causal Inference" and the paper "Necessary and sufficient conditions for causal feature selection in time series with latent common causes" by Mastakouri et al. would be helpful.**
>
> **Response:** Thanks for your feedback. It seems there might be a misunderstanding about the challenges and setup of our problem. In this paper, we study the **Directed Summary Causal Graph on Subsampled Time Series with instantaneous effects using only two time-slices**, which differ from those addressed in the works of Section 10.3 of Peters et al. (2017) and Section 2 of Mastakouri et al. (2021). We appreciate your suggestion and incorporate an additional section to discuss these comparisons, thereby clarifying the novelty of our work.
>
> **Our theorem differs from that of Peters et al. (2017):** Theorems 10.1, 10.2, 10.3, and 10.4 in Section 10 of \citet{peters2017elements} rely on all previous time-slices within the observation windows that are accessible, i.e.,  $\mathbf{X}_{\operatorname{past}(t)}$ could be observed, which are not satisfied in subsampled time-series. In Section 10.2.1, \citet{peters2017elements} also noted that current methods for subsampled time-series require well-modeled interventions. Without well-defined interventions, it remains challenging to learn causal structures from subsampled time series without efficient solutions.
>
> In our Theorem 1, given two time-slice observations $\mathcal{D} = \{ \boldsymbol{X}^{t_a}, \boldsymbol{X}^{t_b} \}_{t_a < t_b}$, $X_j^{t_b}$ is a descendant node of $X_i^{t_b}$ iff $X_i^{t_a} \not \perp  X_j^{t_b} \mid \mathbf{an}_i^{t_a}$. We rigorously prove that it is both sound and complete under some graph constraints, i.e., acyclic summary causal graph, stationary full-time graph, and Markov property.
>
> **Our theorem differs from that of Mastakouri et al. (2021):** Mastakouri et al. (2021) focus on the detection of direct and indirect causes of a given target time series, rather than full graph discovery. Their theory requires that each observed candidate time series must be a non-descendant node of the target time series, and they need to conduct two conditional independence tests for each observed candidate to identify the direct causes of the target. In contrast, our theory not only constructs the Descendant Hierarchical Topology of the Directed Summary Causal Graph but also requires only one conditional independence test per component. Furthermore, the full graph studied by Mastakouri et al. (2021) does not include instantaneous effects.
>
> > **[Q6 & W4] Equation (1) is more restrictive than an additive noise model. An additive noise model is defined as Y = f(instantaneous parents, all other lagged parents) + N in a time series setting. In your case, you restrict the lagged variables to be separate functions and has only an additive influence. This seems quite restrictive.**
>
> **Response:**  Thanks for your insightful question. Our proposed theory and the DHT-CIT algorithm do not rely on additional structural or distributional assumptions, except for the faithfulness assumption. The causal structure-function (Eq. (1)) can be directly reformulated to the additive noise model $X_i^{\tau}=f_i (\mathbf{pa}_i^\tau, X_i^{\tau-1}, \mathbf{pa}_i^{\tau-1} )+\epsilon_i^{\tau}$ or more general function $X_i^\tau=f_i (\mathbf{pa}_i^\tau, X_i^{\tau-1}, \mathbf{pa}_i^{\tau-1}, \epsilon_i^\tau)$, where $f_i(\cdot)$ is a twice continuously differentiable function capturing the instantaneous effects from its parents $\mathbf{pa}_i^\tau$ at time $\tau$ and a non-zero time-lagged effect from variable $X_i^{\tau-1}$ at time $\tau-1$. **We have revised the corresponding description in the Problem Setup and Experiments sections.**

---

> ### Author Response · Authors · 2023-11-17
> **Responses by Authors (Part 5): Clarifying Assumptions and Definitions**
>
> > **[Q7] You refer to "unique orderings" of the topology. However, this appears impossible since equivalent orderings exist (e.g. X → Y ← Z has completely equivalent orderings (X,Z,Y) and (Z,X,Y)). If the "uniqueness" refers to something else, then this should be clarified early on.**
>
> **Response:** **The descendant hierarchical topology is unique and contains fewer non-essential edges when compared to non-unique complete topological ordering.** In the descendant hierarchical topology, each node $X_i^t$ establishes direct edges pointing to its descendant nodes $\mathbf{de}_i^t$. In the case of X → Y ← Z, it becomes evident that only two descendant relationships can be established: de(X) = Y and de(Z) = Y. These relationships correspond to two direct edges, X → Y and Z → Y, within the unique descendant hierarchical topology (X → Y ← Z).
>
> **The complete topological orderings of summary graph are non-unique and may contains numerous spurious edges:** The orderings (X,Z,Y) and (Z,X,Y) are complete topological orderings rather than our descendant hierarchical topology proposed in this paper.
>
>
> > **[Q8] Assumption 1 seems flawed. X^t should be independent of X^t-i-k given X^t-1, ..., X^t-i since any previous lag could directly impact the current time lag (skip connection), rendering them dependent.**
>
> **Response:**  **Assumption 1 is the strict definition of the first-order Markov property for time series.** It posits that the future state ($\boldsymbol{X}^{t+1}$) depends exclusively on the current state ($\boldsymbol{X}^{t}$), disregarding any influence from earlier history ($\boldsymbol{X}^{1 \cdots t-1}$). This definition precisely captures the essence of a first-order Markov process, where only the immediate past state is relevant for predicting the future state.
>
> Furthermore, **we can relax the first-order Markov Assumption to a high-order Markov Assumption**. This means that the future time-slice $\boldsymbol{X}^{t+1}$ depends only on states $\boldsymbol{X}^{t \cdots t-q+1}$ and does not directly depend on states $\boldsymbol{X}^{1 \cdots t-q}$. Then, with $q+1$ time-slices ($\boldsymbol{X}^{t_a \cdots t_a-q+1}$ and $\boldsymbol{X}^{t_b}$), we can use $\mathbf{an}^{t_a \cdots t_a-q+1}$ to replace the condition set $\mathbf{an}^{t_a}$ to infer the Descendant-oriented Conditional Independence Criteria (Theorem 1). However, in this paper, we focus exclusively on the two-time-slices algorithm to demonstrate our theorem and algorithm. **We have included a section to discuss the extension of the Markov Assumption to a high-order Markov Assumption in Appendix F.**
>
> > **[Q9] Definition 2 is unclear. What about X → Y → Z? It has 3 layers (X)_L1, (Y)_L2, (Z)_L3. By the definition, L1 > L3, so X has a direct edge to Z, which would be wrong?**
>
> **Response:**  We have to emphasize the definition of Hierarchical Topological Ordering: If there is a directed edge from $X_{\pi_i}$ to $X_{\pi_j}$, then $l_{\pi_i} > l_{\pi_j}$. The statement "$X_{\pi_i} \rightarrow X_{\pi_j}$" is a sufficient but not necessary condition of "$l_{\pi_i} > l_{\pi_j}$". By the definition, X has a direct edge to Y and Y has a direct edge to Z, so $l_X > l_Y > l_Z$.
>
> > **[Q10] When you mention 'DAG', do you refer to the summary graph? This should be clarified.**
>
> **Response:** Thanks for your feedback. In this paper, 'DAG' refers to the directed acyclic graph of the summary graph in Figure 1(a), and we have clarified this in the main text.

---

> > ### Author Response · Authors · 2023-11-17
> > **Responses by Authors (Part 6): Edge pruning problem, Time series methods, Arbitrary non-linear connections experiments**
> >
> > > **[Q11] After the introduction, it is unclear why the problem isn't an edge pruning problem, as mentioned later. It is unclear why the 'naive' approach of pruning edges after obtaining the causal order is insufficient.**
> >
> > **Response:** As Reviewer w8o3 pointed out in his comment, the problem studied in this paper can indeed be regarded as an edge-pruning problem.  The proposed DHT-CIT method is a model-free plugin module that can be **incorporated into the Complete Topological Ordering learned by SCORE to remove the unnecessary edges to improve topological ordering.**
> >
> > **The 'naive' method of pruning edges is not only a high-complexity traversal algorithm but also suffers from reduced accuracy as unnecessary edges increases.** In topology-based methods, SCORE often generates a complete topological ordering with $d(d-1)/2$ edges, where $d$ denotes the number of nodes. However, many of these edges are unnecessary and point to non-descendants, requiring pruning, which can decrease the accuracy and efficiency of the pruning process. Our DHT-CIT algorithm improves upon existing typology-based methods by eliminating numerous spurious edges in the learned Descendant Hierarchical Topology, thus **effectively reducing the search space in the pruning stage and enhancing the accuracy of the learned DAGs.**
> >
> > > **[Q12] While the experiments have many approaches in the comparison and different data set sizes, more relevant comparisons to time series methods like Granger causality or VARLiNGAM would be insightful. The functional relationships could also be more general, like neural networks with random weights to generate arbitrary non-linear connections.**
> >
> > **Response:**  Thank you for your suggestion. We have incorporated CD-NOD [Huang et al., 2020], Granger causality [Granger et al., 1969; Shojaie et al., 2010] and VARLiNGAM [Hyvärinen et al., 2010] as baselines in our main experiments. For Granger causality and VARLiNGAM, we use the latest implementation from the causal-learn package. As we have analyzed, traditional time series algorithms Granger and VARLiNGAM rely on the modeling of the standard measurement timescale and require the absence of latent time slices. Therefore, under the subsampling time series setting in this paper, they perform poorly in identifying the causal graph, and even underperform compared to temporal variants of the conventional method designed for non-temporal data. While the CD-NOD method shows promising results in small-size datasets like sin-10-10, its computational cost increases super-exponentially with the growth of the graph size, and its performance is also noticeably inferior to our method.
> >
> > The functional relationships generated by neural networks with random weights **might break the causal Markov condition and faithfulness assumption**, so we didn't conduct that experiment in this paper. Instead, we have thoroughly explored **varying time-lagged edges and denser graphs in Figure 2** and included experiments with other complex **non-linear relationships (such as Sin, Sigmoid, and Poly functions) in Table 4 in the Appendix**. The application of our algorithm to the real PM-CMR dataset further showcases its superior performance and scalability in handling larger and denser graphs. We believe these experiments adequately demonstrate the efficacy of our method.

---

> ### Author Response · Authors · 2023-11-17
> **Responses by Authors (Part 7): Other minor problems**
>
> > **[Q13] You mention causal discovery algorithms identify the equivalence class, which is true for some (e.g. PC, FCI) but not all (e.g. NOTEARS aims to recover the whole DAG).**
>
> **Response:**  Thank you for the correction. Most constraint-based methods (e.g. PC, FCI) typically find causal structures within an equivalence class, resulting in a limited understanding of the underlying causal relationships. Score-based methods (e.g. NOTEARS, GraNDAG) rely on local heuristics to enforce acyclicity constraints, which can be insufficient for effectively handling large datasets. Additionally, the causal graphs produced by minimizing a specific score function are not guaranteed to be entirely accurate. We have revised it in the revision.
>
>
> > **[Q14] The time slice definition is confusing. You mention an initial slice from 1 to t-1 and the present is t. Where is the gap between them?**
>
> **Response:**  In this context, **a time slice denotes the cross-sectional observational data** on units captured at a specific time point or time instant (within the given measurement accuracy) on the timeline of an event or process. 'Limited time slices' means that we can only access a few cross-sectional pieces of the samples at limited time points, which might be consecutive moments or non-consecutive moments.
>
> In this paper, the 'initial record' and 'present record' are used in the healthcare example (Figure 1(b)), representing the patient's records from the first visit and second visit, respectively. Then, doctors typically compare the patient’s second follow-up visits to assess the patient's condition for more precise treatment. For further clarify, we would use $\mathcal{D} = \{ \boldsymbol{X}^{t_a}, \boldsymbol{X}^{t_b} \}_{t_a < t_b}$ to denote two time-slices, where $\boldsymbol{X}^{t_a}$ and $\boldsymbol{X}^{t_b}$ are observations of some system at time $t_a$ and $t_b$ ($1 < t_a < t_b$).
>
> > **[Q15] Unclear why a time slice is an intervention. It seems a time slice just means recorded at different times (i.e. with a time gap)?**
>
> **Response:** As indicated by red marking in Figure 1(b), an intervention on the variable $X_2^{t-3}$ would transfer its effect to descendants ($X_3^{t-3}, X_4^{t-3}$), but not to non-descendants ($X_1^{t-3}, X_5^{t-3}$). This helps in swiftly identifying each variable's descendants, enhancing topological ordering. Inspired by this concept, as illustrated in Fig. 1(b), where each node is influenced by its ancestors and itself, each variable in the earlier time-slice would transmit self-perturbations to both itself and its descendants in the subsequent time-slice, serving as simulated interventions.
>
> > **[Q16] Consider avoiding the use of notations that has not yet been introduced, such as $X_i^{t-3}$ in the introduction.**
>
> **Response:** Thanks for your advice. As shown in Figure 1(b), $X_i^{t-3}$ denotes the observations of i-th node $X_i$ at time $t-3$.  We have added the corresponding description in the caption of Figure 1.
>
> > **[Q17] You write "two time-slides data" at the end of the introduction. What is a "slide" here?**
>
> **Response:** This is a typo: "two time-slides data" → "two time-slices data".
>
> > **[Q18] Defining X_i as a random variable does not make sense in the time series setting, where X_i^t is a random variable (with typically only one observation). See Section 10 of "Elements of Causal Inference".**
>
> **Response:** Thanks for your comment. In this paper, variable $X_i^t$ represents a measurement of the $i$-th observable $X_i$ of units at time $t$, where observable $X_i$ denotes the $i$-th feature or attribute of multiple samples in time series. We have revised it in the revision.
>
> ----
>
> **According to your comments, our paper has been significantly improved in areas such as the definition of the problem setup, positioning in related work, discussions on baselines, theorem, and data generation function. We have uploaded the revised version of our paper (with track changes marked in blue). We welcome any further technical advice or questions on this work and we will do our best to address your concerns.**

---

> > ### Comment · Reviewer_mehL · 2023-11-20
> >
> > I want to thank the authors for their detailed responses. Some of my concerns were addressed, but I still think there are too many smaller details that need more careful revision. For instance, seeing X_i^t as a sample is confusing, since typically each single time lag t for a time series X_i is a random variable itself from which you obtain a single sample. Therefore, X_i^t should be a random variable, from which you obtain one observation. I referenced Section 10 of the book "Elements of Causal Inference" to give an example how it could be formalized cleaner. You also mentioned that you have the special case of having instantaneous effects, but also this is discussed in the book (see e.g., Figure 10.5).
> >
> > That being said, I appreciate the authors effort in improving the paper and I am willing to increase my score. Nevertheless, I still think that the paper requires additional changes.

---

> ### Author Response · Authors · 2023-11-21
> **Thankful for Your Revised Scoring and Opportunity to Further Clarify**
>
> Dear reviewer,
>
> Thank you for raising the score. We appreciate your detailed feedback and understand your concerns regarding $X_i^t$ and the problem setup. However, **there might still be some misunderstandings about these two aspects**, and we are willing to explain and clarify them for you again.
>
> **The definition of $X_i^t$ in Problem Setup Section**: "Let $\boldsymbol{X} = \\{X_i^{\tau}\\}_{d \times t}$ denote the full $d$-variate time series with all time slices $\boldsymbol{X}^{\tau}$ at $t$ time points, where ${X}_i^{\tau}$, $i \in \\{1,2,\cdots,d\\}$ and $\tau \in \\{1,2,\cdots,t\\}$, is a random vector comprising observations of $n$ samples. For simplicity of notation, we will not discuss each sample individually. Instead, we refer to ${X}_i^{\tau}$ as a random variable/vector when discussing the causal structure", which has been highlighted in blue in the updated revision. **In the text, we never view $X_i^t$ as a sample, as it is a random vector containing observations of $n$ samples/repetitions. We have revised all corresponding descriptions about ${X}_i^{\tau}$, $\boldsymbol{X}_i$, and $\boldsymbol{X}^{\tau}$ in the last updated revision.**
>
> **As for the cleaner definition of the problem-setting**: "This paper studies the summary causal graph on subsampled time series with instantaneous effects using only two time-slices, in which time-slices (cross-sectional observations) are sampled at a coarser timescale than the causal timescale of the underlying system. Given the presence of unmeasured time-slices, conventional causal discovery methods designed for standard time series data would produce significant errors about the system’s causal structure", which has been highlighted in blue in the Abstract and Introduction Section.
>
> In Section 10 of the book "Elements of Causal Inference", the sufficient condition for the identifiability of the direction of instantaneous effects in acyclic summary graphs has been given by Theorem 10.2. However, Theorems 10.1, 10.2, 10.3, and 10.4 in Section 10 of the book rely on all previous time-slices within the observation windows that are accessible, i.e.,  $X_{past(t)}$ could be observed, which are not satisfied in subsampled time-series. **In Section 10.2.1, Peters et al. (2017) also noted that current methods for subsampled time-series require well-modeled interventions. Without well-defined interventions, it remains challenging to learn causal structures from subsampled time series without efficient solutions.** In our Theorem 1, given two time-slice observations $\mathcal{D} = \{ \boldsymbol{X}^{t_a}, \boldsymbol{X}^{t_b} \}_{t_a < t_b}$, $X_j^{t_b}$ is a descendant node of $X_i^{t_b}$ iff $X_i^{t_a} \not \perp  X_j^{t_b} \mid \mathbf{an}_i^{t_a}$. We rigorously prove that it is both sound and complete under some graph constraints, i.e., acyclic summary causal graph, stationary full-time graph, and Markov property. We have revised the corresponding descriptions in the last updated revision.
>
> ----
>
> **Thank you for dedicating time to review our manuscript and providing valuable insights. We have carefully incorporated your suggestions into our revised paper, with changes marked in blue. We hope our responses address your concerns adequately. Please feel free to let us know if you believe the paper needs further changes. We would appreciate the opportunity to address them promptly.**

---

### Official Review · Reviewer_w8o3 · 2023-10-29

**Soundness:** 3 good
**Presentation:** 3 good
**Contribution:** 4 excellent
**Rating:** 8
**Confidence:** 4

**Summary:**

This paper uses time-series data to replace experimental interventions to help identify the true causal DAG. Under the acyclic summary graph, the authors introduce auxiliary instrumental variables in time-series to act as interventions for identifying (non-)descendants of each variables and propose a Descendant Hierarchical Topology for learning DAGs.

**Strengths:**

- This paper presents auxiliary instrumental variables in time-series to improve the topology learning and proposes a novel DHT-CIT algorithm to accurately identify the true DAG.

- The proposed method has the potential to serve as a plugin module to help existing topology-based methods improve the learned DAGs.

- This paper is well-organized and easy-to-follow. This paper provides a comprehensive review of traditional causal discovery methods and topology-based methods.

- The authors perform extensive experiments to demonstrate the effectiveness of the proposed method. The empirical results provide solid evidence to support the claim of this paper.

**Weaknesses:**

- I have some questions: Does this paper rely on Gaussian Models? While the experimental results indicate that the proposed method performs well on some Non-Gaussian Models.

- It would be time-consuming to conduct conditional independence tests for all variables from scratch. Why not use the proposed method directly, based on the Complete Topological Ordering learned by SCORE, to remove the unnecessary edges to identify the Descendant Hierarchical Topology? I think this would save at least half of the time.

- Can the proposed DHT-CIT be regarded as a model-free plugin module that can be incorporated into any existing topology-based method to improve the learned topological graph? Additionally, can it function as a test tool for selecting the true causal graph from Markov equivalence classes?

**Questions:**

See above

---

> ### Author Response · Authors · 2023-11-17
> **Responses by Authors**
>
> Dear Reviewer,
>
> We sincerely appreciate your constructive comments to improve our paper. Below, we address your concerns point by point. Please kindly let us know whether you have any further questions or suggestions.
>
> > **[Q1] Does this paper rely on Gaussian Models? While the experimental results indicate that the proposed method performs well on some Non-Gaussian Models.**
>
> **Response:** Thanks for your question. Our DHT-CIT algorithm does not rely on Gaussian Models. In this paper, we use two time-slices as conditional instrumental variables to simulate the effects of exogenous interventions. When applied to a variable, these simulated interventions only affect the variable's value, and the permutation would propagate to its descendant nodes. Then, we can apply conditional independence tests to capture these intervention-related permutations for identifying each variable's descendants, **without requiring any structural or distributional assumptions about the data**.
>
> > **[Q2] It would be time-consuming to conduct conditional independence tests for all variables from scratch. Why not use the proposed method directly, based on the Complete Topological Ordering learned by SCORE, to remove the unnecessary edges to identify the Descendant Hierarchical Topology? I think this would save at least half of the time.**
>
> **Response:** Thanks for your suggestion. Applying the DHT-CIT algorithm to the Complete Topological Ordering derived from SCORE is indeed an option that may reduce computational time. However, this approach does not improve the identification of the Descendant Hierarchical Topology. Additionally, **SCORE relies on Gaussian Models, but our DHT-CIT algorithm does not require any structural or distributional assumptions about the data**. Our DHT-CIT algorithm is an independent model that is capable of directly learning the descendant hierarchical topology of DAGs from observational data and does not rely on any other topology-based methods.
>
> > **[Q3] Can the proposed DHT-CIT be regarded as a model-free plugin module that can be incorporated into any existing topology-based method to improve the learned topological graph? Additionally, can it function as a test tool for selecting the true causal graph from Markov equivalence classes?**
>
> **Response:** Yes, the proposed DHT-CIT can be **integrated as a module into any existing topology-based method to enhance the topological ordering**. Additionally, our DHT-CIT algorithm is capable of identifying the true causal graph from the Markov equivalence classes that are typically learned using traditional methods. For instance, our DHT-CIT algorithm can effectively **use the Descendant Hierarchical Topology to orient the undirected edges outputted by a constraint-based algorithm** such as PCMCI [Runge et al., 2019]. We have included an additional section to discuss the potential application of our DHT-CIT algorithm in Appendix H.
>
> ----
>
> **Thank you for your constructive feedback which has been invaluable in enhancing our work. We have uploaded the revised version of our paper (with track changes marked in blue). We welcome any further technical advice or questions on this work and we will do our best to address your concerns.**

---

### Official Review · Reviewer_5w1M · 2023-10-30

**Soundness:** 3 good
**Presentation:** 3 good
**Contribution:** 3 good
**Rating:** 6
**Confidence:** 2

**Summary:**

The paper is about learning causal relations from two time-slices data, which are more common and realistic than full interventions. The paper proposes a novel algorithm called DHT-CIT, which uses the previous time-slice as an auxiliary instrumental variable to quickly identify the descendants and non-descendants of each node in the causal graph. DHT-CIT can construct a more precise and unique topological ordering, which reduces the search space and spurious edges for learning the true DAG. The paper provides theoretical proofs, synthetic experiments, and a real-world application to demonstrate the effectiveness of DHT-CIT.

**Strengths:**

1.The paper proposes a novel algorithm, DHT-CIT, that can learn a unique descendant hierarchical topology from two time-slices of data with auxiliary instrumental variables1.
2.The paper shows that DHT-CIT can reduce the search space and address the acyclicity constraint in causal discovery, as well as eliminate numerous spurious edges in the learned topology.
3.The paper demonstrates the superior and robust performance of DHT-CIT on both synthetic and real-world data, compared to several state-of-the-art baselines.

**Weaknesses:**

1.The paper builds on existing methods and the main contribution is the use of instrumental variables to improve the topological ordering, but this idea seems have been explored in previous works.
2.The paper only evaluates the algorithm on synthetic data and one real-world dataset. The synthetic data are generated from linear Gaussian models, which may not reflect the complexity and diversity of real-world data. It would be better to conduct more experiments on real-world data from various fields.

**Questions:**

1.How do you handle the cases where the previous time-slice data is missing or unreliable? How does this affect the performance of your algorithm?
2.What are the advantages and limitations of using two time-slices data?
3.How do you generalize your algorithm to other domains and applications, such as social sciences, economics, or biology? What are the challenges and opportunities for applying your algorithm to these fields?

---

> ### Author Response · Authors · 2023-11-17
> **Responses by Authors (Part 1)**
>
> Dear Reviewer,
>
> We sincerely appreciate your constructive comments, which have been instrumental in enhancing our manuscript. Below, we have provided detailed explanations addressing your concerns and uploaded the revised version of our paper (with track changes marked in blue). We would be very grateful if you could review these responses and inform us of any additional issues or further feedback you may have.
>
>
> > **[W1] The paper builds on existing methods and the main contribution is the use of instrumental variables to improve the topological ordering, but this idea seems have been explored in previous works.**
>
>
> **Response:** While using interventions (i.e., instrumental variables) can improve the topological ordering, **full interventions to obtain instrumental variables are often expensive, unethical, or even infeasible**. Therefore, we explore using **conditional instrumental variables in two time-slices** as a substitute for instrumental variables (interventions) to improve causal ordering in **subsampled time series**.
>
> We have to emphasize that the main contribution of our paper lies in **identifying the causal structure in subsampled time series with instantaneous effects using only two time-slices**, in which time-slices (cross-sectional observations) are sampled at a coarser timescale than the causal timescale of the underlying system. This task presents more challenges compared to the Standard Time Series problem: (1) We only observe two time-slices from the Subsampled Time Series, and numerous unmeasured time slices may be latent in the subsampled time series; (2) We focus on subsampled time series with instantaneous effects, thus, the theorems proved by standard time-series assumption are not available, e.g., Granger causality, Peters et al., (2017) and Mastakouri et al., (2021); (3) We don't rely on full interventions to identify the Directed Summary Causal Graph, as full interventions are often expensive, unethical, or even infeasible. Given the presence of unmeasured time-slices, conventional causal discovery methods designed for standard time series data would produce significant errors about the system’s causal structure. **We are the first to use just two time-slices to identify causal structure of subsampled time series, addressing the issue of missing time-slices in time series**.
>
> We have revised and rephrased the connection between our work and existing methods in the Introduction and Related Work sections, highlighted in blue, to further clarify our paper's position in the field.
>
> > **[W2] The paper only evaluates the algorithm on synthetic data and one real-world dataset. The synthetic data are generated from linear Gaussian models, which may not reflect the complexity and diversity of real-world data. It would be better to conduct more experiments on real-world data from various fields.**
>
> **Response:** Thank you for your feedback. However, it appears there might be a misunderstanding regarding the scope of our experiments. As shown in Tables 1, 2, 4, and Figure 2, **we have conducted extensive non-linear experiments with various noise types, which the reviewer may have overlooked**. In the main text, similar to the experimental setups of previous studies (Rolland et al., 2022; Sanchez et al., 2022; Montagna et al., 2023), we have conducted comprehensive experiments to demonstrate the superiority of our DHT-CIT algorithm. To better reflect the complexity and diversity of real-world data, we generated multiple synthetic datasets not only from additive **non-linear Gaussian models (Table 1)** but also from **non-linear Laplace models and non-linear Uniform models (Table 2)**. Additionally, we explored **varying time-lagged edges and denser graphs in Figure 2** and included experiments with other complex **non-linear relationships (such as Sin, Sigmoid, and Poly functions) in Table 4 in the Appendix**. The application of our algorithm to the real PM-CMR dataset further showcases its superior performance and scalability in handling larger and denser graphs. We believe these experiments adequately demonstrate the efficacy of our method. While we are open to conducting more experiments with real-world data from various fields, obtaining such data presents challenges, and their use in evaluating baselines is limited due to the absence of known real causal graphs.

---

> ### Author Response · Authors · 2023-11-17
> **Responses by Authors (Part 2)**
>
> > **[Q1] How do you handle the cases where the previous time-slice data is missing or unreliable? How does this affect the performance of your algorithm?**
>
> **Response:** Thank you for your question, which highlights the core motivation and main contribution of this paper: we identify the summary causal graph from subsampled time series with instantaneous effects using only limited time-slices, in which numerous previous time-slices are missing or unreliable.
>
> In many applications, the time series sampling process may be slower than the timescale of causal processes, resulting in numerous previous time-slices are missing or unreliable (**Problem**). In the presence of unmeasured time-slices, relying solely on a single time-slice is insufficient for identifying causal relations (**Challenges**). Therefore, our **Motivation** is to use just two reliable time-slices to explore the summary causal graph of subsampled time series, rather than depending on all previous time-slices that are available and reliable (**the limitation of traditional methods**). In this paper, we demonstrate that if two valid time slices at two arbitrary moments are available, the variables in the earlier slice can be used as conditional instrumental variables to replace interventions and improve topological ordering. This method significantly relaxes the assumption inherent in traditional time series studies that depend on modeling causal structures at the system timescale, causal sufficiency, and all time slices in the observation windows could be observed (**our contribution**)[Granger, 1969, 1980; Lutkepohl, 2005; Hyvarinen et al., 2010; Luo et al., 2015; Nauta et al., 2019; Runge et al., 2019; Runge et al., 2020; Bussmann et al., 2021; Lowe et al., 2022; Assaad et al., 2022]. However, it is important to note that **if no previous time-slice data is available or reliable, our approach, like other causal discovery algorithms, will not produce identifiable results.**
>
> > **[Q2] What are the advantages and limitations of using two time-slices data?**
>
> **Response:** We have added an additional section to discuss the advantages and limitations of using two time-slices data in the Appendix G.
>
> **Advantage:** **(1) Enhanced Topological Ordering**: Traditional topology-based methods typically produce non-unique topological orderings with numerous spurious edges, resulting in decreased accuracy and efficiency in downstream search tasks. By using two time-slices as auxiliary instrumental variables, we can learn causal relations more efficiently, with a reduced search space and fewer spurious edges. **(2) Feasibility in Intervention-Limited Contexts**: Using interventional data can quickly identify (non-)descendants for each node and construct a more precise topological ordering. In scenarios where interventions are infeasible, unethical, or too costly, using two time-slices to replace intervention can be a practical alternative. **(3) Reduced Data Requirements**: In time series scenarios, traditional methods depend on the modeling causal structures at the system timescale, causal sufficiency, and all time slices in the observation windows could be observed. In this paper, we propose exploring limited time-slices, i.e., two reliable time-slices, to ease the data requirements. (4)  **Scaling to Non-linear and Non-Gaussian Models**: We use two time-slices as conditional instrumental variables to simulate exogenous interventions. When applied to a variable, these simulated interventions only affect the variable's value, and the permutation would propagate to its descendant nodes. Then, we can apply conditional independence tests to capture these intervention-related permutations for identifying each variable's descendants, without requiring any structural or distributional assumptions about the data.
>
> **Limitations:**  Our DHT-CIT algorithm strictly relies on the **Acyclic Summary Causal Graph and Consistency Throughout Time Assumptions**, which are common in time series studies [Assaad et al., 2022]. Additionally, if no previous time-slice data is available or reliable, both the two time-slices approach and other causal discovery algorithms will not yield identifiable results.
>
> Notably, in this paper, **we can relax the Markov Assumption to a high-order Markov Assumption**. This means that the future time-slice $\boldsymbol{X}^{t+1}$ depends only on states $\boldsymbol{X}^{t \cdots t-q+1}$ and does not directly depend on states $\boldsymbol{X}^{1 \cdots t-q}$. Then, with $q+1$ time-slices ($\boldsymbol{X}^{t_a \cdots t_a-q+1}$ and $\boldsymbol{X}^{t_b}$), we can use $\mathbf{an}^{t_a \cdots t_a-q+1}$ to replace the condition set $\mathbf{an}^{t_a}$ to infer the Descendant-oriented Conditional Independence Criteria (Theorem 1). However, in this paper, we focus exclusively on the two-time-slices algorithm to demonstrate our theorem and algorithm. We have added a section to discuss the high-order Markov Assumption in Appendix F.

---

> ### Author Response · Authors · 2023-11-17
> **Responses by Authors (Part 3)**
>
> > **[Q3] How do you generalize your algorithm to other domains and applications, such as social sciences, economics, or biology? What are the challenges and opportunities for applying your algorithm to these fields?**
>
> **Response:** As long as the common time series causal assumptions in [Assaad et al., 2022] and the q-order Markov Assumption are satisfied, we can directly extend our algorithm to other domains and applications with q+1 time-slices ($\boldsymbol{X}^{t_a \cdots t_a-q+1}$ and $\boldsymbol{X}^{t_b}$). For example, we can analyze the causal graph of city status variables in PM-CMR [Wyatt et al., 2020], and explore the relationships between various factors affecting soil moisture. **In human genomics and gene expression, we also can establish two-time-slices causal relationships (surjections: where each expression variable can find a corresponding conditional instrumental variable in the genomic sequence variables).** The challenges arise as different time series data may adhere to various high-order Markov Assumptions, which we need to identify. Additionally, sometimes the temporal transfer of events/processes might conceal causal relationships, requiring further extraction, such as the two-time-slices causal relationships between genomic and gene expression data. The DHT-CIT algorithm provides an excellent tool and opportunity for identifying the topological ordering in the aforementioned forms of data. We do not present experiments for all fields within this paper, as data acquisition is an expensive process. We have included an additional section to discuss the future application in Appendix H, and if we have any updates on the above datasets, we will post these updates on our project pages.
>
> ----
>
> **According to your comments, our paper has undergone significant improvements regarding the motivation, problem setup, differentiation from traditional methods, our contribution, and its potential application to real-world scenarios. We have uploaded the revised version of our paper (with track changes marked in blue). We welcome any further technical advice or questions on this work and we will make our best to address your concerns.**

---

> ### Author Response · Authors · 2023-11-21
> **Looking Forward to Your Further Feedback**
>
> Dear Reviewer,
>
> Thank you for dedicating your time to reviewing our manuscript and offering valuable insights. We look forward to any further feedback you may provide.
>
> In response to your constructive feedback, we have made significant revisions to our manuscriptt. We have clarified the problem setting, established a clearer positioning for our work, elaborated on the advantages and limitations of using two time-slices in subsampled time series data, and discussed potential real-world applications in more detail. These changes, marked in blue for your convenience, aim to address the concerns you raised in your previous review. Provided that your concerns have been well-addressed, we would greatly appreciate it if you would consider raising your score.
>
> Should you have any further comments or queries, we would greatly appreciate the opportunity to address them promptly.
>
> Warm regards,
>
> Authors

---

> > ### Comment · Reviewer_5w1M · 2023-12-01
> >
> > Thank you for the prompt responses and the efforts you put into addressing my questions about your paper. After carefully reviewing your explanations, I acknowledged your clarification on the issue related to the scope of your experiments and the relevant experimental results. I appreciate the clarity you've brought to the matter, and it has indeed provided me with a clearer perspective on the advantages and limitations of using two time-slices data. Therefore, I have decided to revise my initial rating score and have adjusted it to 6.

---

### Author Response · Authors · 2023-11-17
**Question: Can we make changes to the title and abstract in the updated reversion?**

Dear Reviewers, Area Chairs, and Program Chairs,

I am writing to inquire **whether it is permissible to revise the title and abstract of our paper in the updated reversion**. The revised title and abstract in this updated version aim to provide a more precise definition of the problem setting and a clearer positioning of our work, while still  staying close to the original submission.

**If such modifications are not permitted, we will revert to the title and abstract of the original submission in our updated revision.**

After carefully considering the comments from the reviewers, I have realized that our munuscript requires a much cleaner definition of the problem setting. Therefore, I kindly request a modification of the title from **"Two Time-Slices Help Topological Ordering for Learning Directed Acyclic Graphs"** to **"Learning Causal Relations from Subsampled Time-Series with only Two Time-Slices"**. I believe that this new title better reflects the content of my paper and will make it more easily understandable to readers.

**The revised abstract would be:**
This paper studies the summary causal graph on subsampled time series with instantaneous effects using only two time-slices, in which time-slices (cross-sectional observations) are sampled at a coarser timescale than the causal timescale of the underlying system. Given the presence of unmeasured time-slices, conventional causal discovery methods designed for standard time series data would produce significant errors about the system’s causal structure. To address these issues, one promising approach is to construct a topological ordering and then prune unnecessary edges to approximate the true causal structure. Existing topology-based methods often yield non-unique orderings with many spurious edges, reducing accuracy and efficiency in downstream search tasks, while using interventional data for more precise ordering is frequently costly, unethical, or infeasible. Therefore, we explore how the more readily available two time-slices data can replace intervention data to improve topological ordering. Based on a conditional independence criterion using two time-slices as auxiliary instrumental variables, we propose a novel Descendant Hierarchical Topology algorithm with Conditional Independence Test (DHT-CIT) to more efficiently learn causal relations in subsampled time series data. Empirical results on both synthetic and real-world datasets demonstrate the superiority of our DHT-CIT algorithm.

Best regards,

Authors

---

### Author Response · Authors · 2023-11-17
**Summary of Changes**

Dear Reviewers, Area Chairs, and Program Chairs,

We thank all reviewers for their great efforts and time placed on reviewing our paper. We have uploaded a revised manuscript based on the reviewers’ feedback, and have highlighted changes from the original submission in blue.

**Here, we summarize the motivation, novelty, and contributions of this work again:**

In many applications, the time series sampling process may be slower than the timescale of causal processes, resulting in numerous previous time-slices (cross-sectional observations) being missing or unreliable (**Problem**). Given the presence of unmeasured time-slices, conventional causal discovery methods designed for standard time series data would produce significant errors about the system’s causal structure (**The limitation of traditional methods: they rely on all time slices in the observation windows could be observed**). Therefore, in subsampled time series with instantaneous effects, in which observations are sampled at a coarser timescale than the causal timescale of the underlying system, our **Motivation and Novelty** is to use just two reliable time-slices to explore the summary causal graph of subsampled time series. **Theoretically**, we demonstrate that if two valid time slices at two arbitrary moments are available, the variables in the earlier slice can be used as conditional instrumental variables to replace interventions and improve topological ordering. Our theorem differs from that of Peters et al. (2017) and Mastakouri et al. (2021). We rigorously prove that it is both sound and complete under some graph constraints, i.e., acyclic summary causal graph, stationary full-time graph, lagged structure and Markov property. The proposed DHT-CIT algorithm significantly relaxes the assumption inherent in traditional time series studies that depend on modeling causal structures at the system timescale, causal sufficiency, and all time slices in the observation windows could be observed (**Contribution**) [Granger, 1969, 1980; Lutkepohl, 2005; Hyvarinen et al., 2010; Luo et al., 2015; Nauta et al., 2019; Runge et al., 2019; Runge et al., 2020; Bussmann et al., 2021; Lowe et al., 2022; Assaad et al., 2022].

**We have updated the paper accordingly, with the changes highlighted in blue:**

- **In Introduction Section**,  we further clarify our paper's position in the field and provide a much cleaner definition of the problem setting: Learning Causal Relations from Subsampled Time Series with instantaneous effects using only two time-slices.
- **In Related Work Section**,  we update a lot of related works about subsampled and standard time series data. We split the **Original Related Work Section** into two parts: one focusing on temporal data is included in the main text, while the other, dealing with non-temporal data, is positioned in Appendix A.
- **In Problem Setup Section**, we reformulate the structural function to the additive noise model $X_i^{\tau}=f_i(\mathbf{pa}_i^{\tau}, X_i^{\tau-1}, \mathbf{pa}_i^{\tau-1})+\epsilon_i^{\tau}$, and update the description about the two time-slices data, while still staying close to the original submission.
- **In Experiment Section**, we revise the description of data generation and highlight the scope of our experiments. Due to limited space, we delete two poor-performed baselines GES and GraNDAG, and then add two traditional time series algorithms Granger and VARLiNGAM.
- **In Conclusion Section**, we provide a summary of our work.
- **In Appendix A**, we move the original related work on non-temporal data to the appendix.
- **In Appendix B**, we discuss the difference between our theorems with traditional methods.
- **In Appendix C**, we discuss the rationale behind the chosen baselines.
- **In Appendix F**, we relax the Markov Assumption to High-Order Markov Models and extend our DHT-CIT algorithm to subsampled time series setting with high-order lagged effect.
- **In Appendix G**, we list the advantages and limitations of using two time-slices data.
- **In Appendix H**, we discuss the potential future applications of our work.

---

### Author Response · Authors · 2023-11-22
**Thank you for your insightful feedback!**

Thank you again for your valuable time and insightful comments! We understand that the concerns are mainly the formulation of two-time-slices in subsampled time series problem, its position in the related works, as well as the novelty, contributions, and applications of our algorithm. There might be some inaccuracies and omissions in our previous statements, and we apologize for any misunderstanding this may have caused. We have included further clarification and discussions in the updated manuscript and our rebuttal to address your concerns, and we sincerely look forward to your reply. Any guidance to where our revision and rebuttal are lacking would be highly appreciated.

With the discussion deadline approaching, we are anticipating any further advice, inquiries or remaining concerns you may have. We will make our best effort to handle them.

---

### Meta-Review · Program_Chairs · 2023-12-11

**Metareview:**

This paper proposes what is called Descendant Hierarchical Topology algorithm with Conditional Independence Test (DHT-CIT) to learn causal relations in subsampled time series data. Empirical results suggest the benefits of this approach. The comparisons include a range of alternatives, particularly on synthetic data, while real data results are more limited. Overall the presentation is reasonable, even though it has not excited all those involved in the reviewing process at first.

PC/SAC Comment: After calibration and downweighting inflated and non-informative reviews, the decision is to reject at this time.  There are significant concerns on positioning and novelty.

**Justification For Why Not Higher Score:**

Please see meta-review

**Justification For Why Not Lower Score:**

N/A

---

### Decision · Program_Chairs · 2024-01-16

Reject